# Mind Your Margin and Boundary: Are Your Distilled Datasets Truly Robust?

**Muquan Li** [1]  **Yingyi Ma** [1]  **Yihong Huang** [1]  **Hang Gou** [1]  **Ke Qin** [1]  **Ming Li** [2]  **Yuan-Fang Li** [3]  **Tao He** [1]

## Abstract

Dataset distillation (DD) compresses a large training set into a small synthetic set for efficient training, but most DD methods optimize only clean accuracy and leave robustness uncontrolled. Recent robust DD methods improve robustness, yet they often suffer from a poor accuracy–robustness trade-off because they (i) treat all adversarially perturbed examples uniformly, despite robust risk being dominated by near-zero robust margins, and (ii) do not explicitly increase inter-class separation in the decision boundary where attacks concentrate. We present **Contrastive Curriculum for Robust Dataset Distillation (C²R)**, a framework that couples an attack-aware curriculum with a contrastive robustness objective. From a robust-margin perspective, we derive a *perturbation score* that approximates each sample's robust hinge, enabling a curriculum that prioritizes the smallest-margin adversaries that most directly drive robust error. In parallel, a class-balanced contrastive robustness loss enforces adversarial invariance while explicitly widening boundary separation across classes. Experiments on CIFAR-10/100, Tiny-ImageNet, and multiple ImageNet-1K subsets under six attacks show that C²R achieves the best robust accuracy, outperforming prior robust DD by $2.8\%$ on average. Code is available at: https://github.com/SLGSP/CCR.

## 1. Introduction

Modern deep learning is widely applied in various scenarios, such as intelligent transportation (Yan et al., 2025), privacy

[1]The Laboratory of Intelligent Collaborative Computing of UESTC, Chengdu, China [2]Guangdong Laboratory of Artificial Intelligence and Digital Economy (SZ), Shenzhen, China [3]Monash University, Melbourne, Australia. Correspondence to: Tao He <tao.he01@hotmail.com>.

*Proceedings of the 43rd International Conference on Machine Learning*, Seoul, South Korea. PMLR 306, 2026. Copyright 2026 by the author(s).

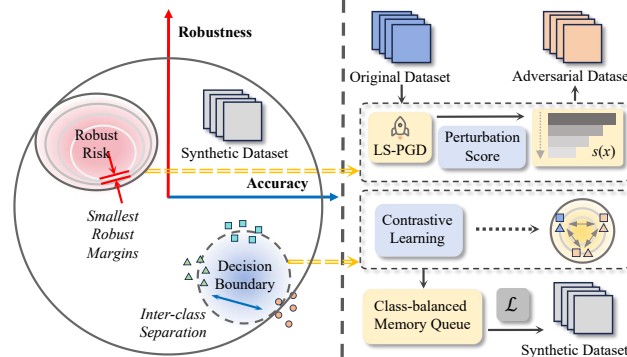

*Figure 1.* Existing robust DD struggles to balance robustness and accuracy by (i) ignoring that robust risk is driven by the smallest margins and (ii) failing to separate classes near decision boundaries. C²R addresses this with an attack-aware curriculum that prioritizes small-margin adversaries and a contrastive robustness loss.

protection (Zhou et al., 2024), and visual understanding (Shen et al., 2026), and is increasingly data-hungry, yet storing and repeatedly training on large-scale corpora can be prohibitive. Dataset Distillation (DD) (Bohdal et al., 2020) addresses this bottleneck by synthesising a small set of training examples such that a model trained on the distilled set approaches the performance of full-data training. Recent progress spans matching-based (Guo et al., 2024; Li et al., 2026b), generative (Wang et al., 2024), parametric (Kim et al., 2022; Liu & Wang, 2023), and decoupled pipelines (Li et al., 2026a), making DD attractive for low-resource training (Chen et al., 2025), continual learning (Masarczyk & Tautkute, 2020; Gu et al., 2024), and privacy-sensitive deployment (Dong et al., 2022).

However, distilled datasets inherit not only the predictive signal of the original data but also its vulnerabilities. In particular, models trained on distilled data can remain highly susceptible to adversarial perturbations (Wu et al., 2025), which is especially concerning when DD is used for deployment in constrained or safety-critical settings. This has motivated emerging work on *robust dataset distillation*. Early approaches (Tsilivis et al., 2022) incorporate robustness-aware objectives via meta-learning, GUARD (Xue et al., 2025) improves stability through curvature regularization, and ROME (Zhou et al., 2025) aligns clean and adversarial feature distributions under an information-bottleneck view. Collectively, these studies suggest robust DD is feasible,

but they also surface open challenges: (1) *principled objectives* that target decision-critical robustness rather than average behavior, (2) *attack-adequate and clearly specified evaluation* (including strong, standardized attacks), and (3) *efficiency*, since robustness typically requires expensive inner-loop adversarial optimization.

This paper argues that two structural gaps limit current robust DD methods in the accuracy–robustness trade-off (Fig. 1). **(i) Margin misprioritization.** Robust risk is governed by the *worst-case* (smallest) robust margins rather than by typical examples (Schmidt et al., 2018; Tsipras et al., 2019). Yet existing pipelines often treat adversarial instances uniformly during distillation, diluting optimization effort away from the highest-risk (small-margin) cases that dominate robust error. **(ii) Boundary neglect.** Many objectives encourage global similarity between clean and adversarial features (e.g., mean alignment), but robust errors are decided near decision boundaries where neighboring classes compete (Madry et al., 2018; Zhang et al., 2019; Chen et al., 2024). Without explicitly enlarging inter-class separation in these regions, distilled sets can be brittle under strong attacks. These observations raise a concrete question: *Can we design a robust distillation objective that (a) concentrates on smallest-margin adversaries and (b) explicitly increases boundary-level class separation, while keeping the distillation cost practical?*

We answer this question with Contrastive and Curriculum Robust Dataset Distillation ($\mathbf{C^2R}$), a simple yet effective framework that couples an attack-aware curriculum with an instance-level contrastive robustness objective (Fig. 1). Motivated by robust-margin theory (Schmidt et al., 2018; Tsipras et al., 2019), $C^2R$ first computes an attack-aware perturbation score that serves as a proxy for each sample's robust hinge (and hence its robust margin), and uses this score to prioritize optimization on the smallest-margin adversarial examples. Second, to regularize decision-critical regions, $C^2R$ replaces class-mean alignment with a contrastive robustness loss that treats clean-adversarial pairs within the same class as positives and samples from other classes as negatives, encouraging adversarial invariance while increasing effective inter-class separation near boundaries. Finally, to reduce the cost of robust distillation, we introduce a *Line-Search PGD (LS-PGD)* attacker to avoid unnecessary backpropagation during inner-loop attack generation, and employ a class-balanced memory queue to amortize contrastive pair construction. Across our benchmarks and attack suites (including strong standardized attacks under a consistent threat model), $C^2R$ improves the robustness–accuracy trade-off over prior robust DD methods, while keeping computation and memory requirements manageable in practice.

In summary, our contributions are fourfold as below:

- **Robust-margin view of DD.** We formalize robustness in distillation through the dataset's minimum robust margin and provide a theoretical justification linking it to minimizing a max robust-hinge surrogate, motivating margin-aware robust distillation objectives.

- **Attack-aware curriculum (AAC) with efficient inner-loop attacks.** We derive a PGD-based perturbation score that estimates each example's robust hinge and use it to focus distillation updates on smallest-margin adversaries. To control cost, we introduce *LS-PGD*, which reduces attack-time backpropagation while maintaining effective adversarial strength.

- **Contrastive robustness loss (CRL) for boundary regularization.** We propose a contrastive objective that enforces clean-adversarial invariance within class while penalizing rival-class proximity, better matching the boundary-level nature of robust errors. A class-balanced memory queue makes CRL more scalable.

- **Empirical evaluation.** On CIFAR-10/100, Tiny-ImageNet, and several ImageNet-1K subsets with diverse attack types, $C^2R$ consistently improves robust accuracy over prior robust DD baselines, and achieves a stronger robustness-accuracy trade-off.

## 2. Related work

**Dataset Distillation.** Dataset distillation (DD) compresses a large training set into a small set of synthetic samples such that models trained on the synthetic set can approach the performance of training on full data (Zhou et al., 2022; Li et al., 2025). Early DD is formulated as bilevel optimization (Wang et al., 2018), where an inner loop trains on synthetic data and an outer loop updates synthetic samples. Later methods use more objectives: gradient matching aligns update directions (Zhao et al., 2021), trajectory matching extends alignment along optimization paths (Cazenavette et al., 2022), and distribution matching aligns feature statistics between real and synthetic sets (Zhao & Bilen, 2023; Wang et al., 2025).

**Adversarially Robust Dataset Distillation.** Although DD reduces training cost, heavy compression can reduce diversity and effective margins, which may hurt adversarial robustness. Robust DD therefore incorporates robustness objectives during synthesis. Some methods use NTK-informed bilevel meta-learning (Tsilivis et al., 2022), while others cast robust dataset learning as a bilevel program to obtain adversarially robust classifiers (Wu et al., 2022). Related work also targets issues that often co-occur with condensation, such as miscalibration (Zhu et al., 2023) and OOD detection (TrustDD) (Ma et al., 2025). Additional directions include curvature regularization for low-cost robustness (Xue et al., 2025), integrating distributional robustness, e.g., group DRO (Vahidian et al., 2025), and information-bottleneck views

that jointly align accuracy and robustness (ROME) (Zhou et al., 2025). Overall, these works highlight that robust DD results are sensitive to the threat model and evaluation protocol, so clearly specifying attack budgets and configurations is important for fair and reproducible comparisons.

## 3. Preliminaries

**Dataset distillation.** Given a training distribution $\mathcal{D}$, dataset distillation (DD) learns a compact synthetic set $X = \{(x_s, y_s)\}_{s=1}^N$ with $N \ll |\mathcal{D}|$ such that training on $X$ approximates training on $\mathcal{D}$. Let $f_\theta$ be a model and $\ell$ a task loss. A standard bilevel formulation is

$$\min_X \ \mathbb{E}_{(x,y)\sim\mathcal{D}}\big[\ell\big(f_{\theta^\star(X)}(x), y\big)\big] \quad (1)$$

$$\text{s.t. } \theta^\star(X) \in \arg\min_\theta \ \mathbb{E}_{(x,y)\sim X}[\ell(f_\theta(x), y)], \quad (2)$$

where the inner problem trains on $X$ and the outer objective evaluates on $\mathcal{D}$.

**Robust dataset distillation.** Robust DD augments DD so that models trained on $X$ are accurate and resilient to adversarial perturbations. A common design (e.g., ROME (Zhou et al., 2025)) optimizes a weighted sum of a classification loss $\mathcal{L}_{\text{perf}}$ and a robustness-matching term $\mathcal{L}_{\text{rob}}$:

$$\mathcal{L}_{\text{perf}} = \mathbb{E}_{(x,y)\sim X}\big[\text{CE}(f_\theta(x), y)\big], \quad (3)$$

$$\mathcal{L} = (1-\eta)\mathcal{L}_{\text{perf}} + \eta\mathcal{L}_{\text{rob}}, \ \ \eta \in [0,1]. \quad (4)$$

To instantiate $\mathcal{L}_{\text{rob}}$, one typically generates adversarial counterparts $\tilde{x}$ (e.g., via PGD (Madry et al., 2018)) for each synthetic sample and encourages embedding consistency between clean and adversarial examples. A representative choice matches class-conditional embedding means:

$$\mathcal{L}_{\text{rob}} = \sum_{c=1}^C \big\|\mathbb{E}_{x\in X_c}\big[e(x)\big] - \mathbb{E}_{\tilde{x}\in\tilde{X}_c}\big[e(\tilde{x})\big]\big\|_2^2, \quad (5)$$

where $e(\cdot)$ is the embedding output, and $X_c$ (resp., $\tilde{X}_c$) are clean (resp., adversarial) synthetic samples of class $c$.

**Limitations motivating our approach.** This formulation makes two simplifying approximations that can weaken robustness. (i) It treats all adversarial companions uniformly, while robust error is dominated by the *worst-case* (small-margin) examples; averaging can under-emphasize the hardest attacks. (ii) Mean matching collapses intra-class geometry in embedding space, potentially masking vulnerable submodes and reducing the ability to preserve fine-grained structure. These observations motivate objectives that (a) emphasize hard adversarial cases and (b) maintain instance-level structure under perturbations.

## 4. Method

Fig. 2 summarizes $C^2R$ for robust dataset distillation. Our design is guided by the margin analysis in Sec. 4.1, which shows that robust risk is dominated by the *minimum* robust margin. $C^2R$ has two components. First, an *Attack-Aware Curriculum* (AAC) prioritizes adversarial companions with small robust margins, so optimization focuses on the low-margin tail that controls robust error (Sec. 4.2). For efficiency, we estimate margins using a warm-started line-search attacker (LS-PGD) that preserves attack strength under a fixed compute budget. Second, CRL replaces class-mean robustness alignment with supervised contrast, pulling each sample toward its adversarial counterpart while pushing it away from competing-class negatives (Sec. 4.3).

### 4.1. Theoretical Insights: Robust Margin as the Decision-Critical Quantity

Adversarial robustness is inherently a worst-case property: performance is determined by whether any allowable perturbation can flip the prediction. This suggests that, in robust DD, the decision-critical statistic is the minimum robust margin over the training set, rather than an average alignment between clean and adversarial representations.

**Robust margin and robust risk.** Let $f_\theta : \mathbb{R}^d \to \mathbb{R}^K$ be a classifier with logits $\{f_k(x)\}_{k=1}^K$. For a labeled example $(x,y)$, define the logit margin $g_\theta(x) = f_y(x) - \max_{k\neq y} f_k(x)$. Given an $\ell_p$-bounded threat set $\Delta = \{\delta : \|\delta\|_p \leq \varepsilon\}$, the robust margin is

$$\underline{m}(x;\theta) = \min_{\delta\in\Delta} g_\theta(x+\delta). \quad (6)$$

An example is robustly classified iff $\underline{m}(x;\theta) > 0$, hence the robust classification risk is $\mathcal{R}_{\text{rob}}(\theta) = \Pr[\underline{m}(x;\theta) \leq 0]$. This makes explicit that robustness depends on the smallest margin achieved within $\Delta$.

**Tractable surrogate and worst-case structure.** Minimizing $\mathcal{R}_{\text{rob}}$ is intractable, so we use the robust hinge surrogate

$$\mathcal{L}_{\text{hinge}}(\theta) = \mathbb{E}_{(x,y)}\Big[[1-\underline{m}(x;\theta)]_+\Big], [z]_+ = \max\{0, z\}. \quad (7)$$

Let $v_i(\theta) = [1 - \underline{m}(x_i;\theta)]_+$ denote the per-sample robust-hinge value. Only samples with $\underline{m}(x_i;\theta) < 1$ contribute non-zero loss, i.e., those close to the decision boundary.

Crucially, since $z \mapsto [1-z]_+$ is monotone non-increasing, the sample that maximizes $v_i$ is exactly the one with the smallest robust margin:

$$\arg\max_{i\in[n]} v_i(\theta) = \arg\min_{i\in[n]} \underline{m}(x_i;\theta), \quad (8)$$

$$\max_{i\in[n]} v_i(\theta) = \big[1 - \min_{i\in[n]} \underline{m}(x_i;\theta)\big]_+. \quad (9)$$

Eqn. 9 formalizes the key point: *improving the worst robust-hinge loss is equal to improving the minimum robust margin.*

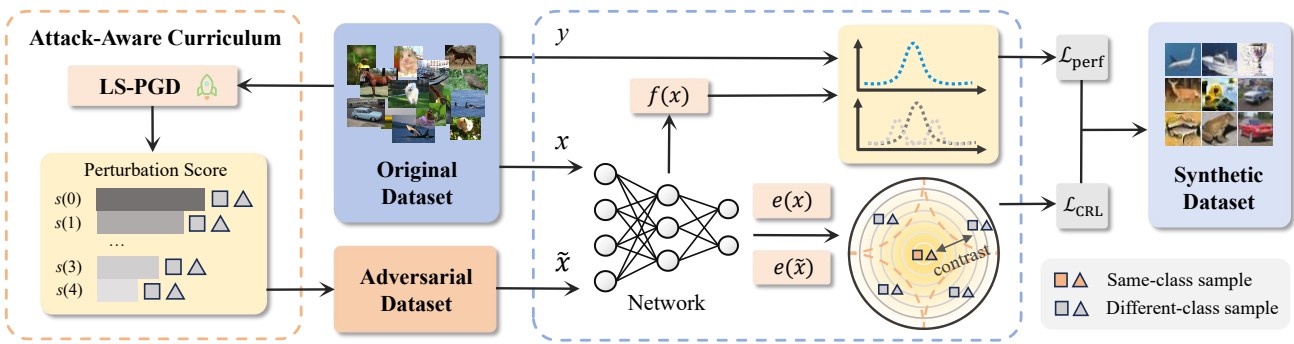

*Figure 2.* **Overview of the proposed C$^2$R framework**. LS-PGD generates adversarial examples and assigns perturbation scores to form an attack-aware curriculum (AAC), while the synthetic dataset is iteratively optimized with a supervised loss and a contrastive robustness loss (CRL) that aligns clean–adversarial features and enlarges the robust decision margin.

Hence, any objective that averages robustness signals across samples can be dominated by many "easy" points and may under-emphasize the few "hard" points that govern $\mathcal{R}_{\mathrm{rob}}$.

**Implication for robust distillation.** In DD, the synthetic set $X$ acts as the training distribution that induces $\theta^\star(X)$. From Eqs. 7–9, robust generalization is driven by the subset of samples with the *smallest* robust margins under $\theta^\star(X)$. This motivates distillation updates that (i) explicitly identify low-$\underline{m}$ (high-$v_i$) examples and (ii) allocate more optimization budget to them, rather than enforcing uniform robustness alignment (e.g., class-mean matching) that can blur the decision-critical tails of the margin distribution.

### 4.2. Attack-Aware Curriculum

The analysis in Sec. 4.1 shows that robust risk is governed by the minimum robust margin. Consequently, robust distillation should devote disproportionate optimization effort to the adversarial examples that attain the smallest margins, rather than treating all adversarial companions uniformly ( Eqn. 5). We instantiate this principle with an *Attack-Aware Curriculum* (AAC) that (i) estimates per-sample robust margins, (ii) uses them to rank attacks by perturbation, and (iii) schedules robustness-alignment updates accordingly.

**Robust-margin estimation via adversarial maximization.** For a synthetic pair $(x, y)$ and threat set $\Delta = \{\delta : \|\delta\|_p \leq \varepsilon\}$, the robust margin is $\underline{m}(x; \theta) = \min_{\delta \in \Delta} g_\theta(x + \delta)$ (Eqn. 6). We approximate the inner minimization using projected gradient descent on the training loss (PGD), producing a strong adversarial companion. With step size $\alpha$, iteration budget $T$, and projection $\Pi_\Delta$, PGD iterates

$$\delta_{t+1} = \Pi_\Delta\Big(\delta_t + \alpha \operatorname{sign}\big(\nabla_x \, \ell(f_\theta(x + \delta_t), y)\big)\Big), \quad (10)$$

and returns $\delta_T$. The resulting margin estimate is

$$\widehat{m}_{\mathrm{rob}}(x; \theta) := g_\theta(x + \delta_T) \approx \underline{m}(x; \theta). \quad (11)$$

This connects the curriculum score directly to the decision-critical quantity in Sec. 4.1.

**LS-PGD: preserving attack strength under a fixed compute budget.** Robust DD repeatedly generates adversarial companions during distillation; naïvely running $T$-step PGD for every sample is costly. We therefore use a warm-started *line-search PGD* (LS-PGD) that reduces backpropagations while maintaining attack strength. For each $x$, we cache the previous perturbation $\hat{\delta}(x)$. If the current loss at $x + \hat{\delta}(x)$ does not decrease, we reuse $\hat{\delta}(x)$. Otherwise, we compute a single ascent direction $v = \operatorname{sign}(\nabla_x \ell(f_\theta(x + \hat{\delta}(x)), y))$ (1 backward pass) and select a step size by forward-only search over a small geometric shortlist $\mathcal{S} = \{\eta_q = \alpha\beta^q\}_{q=0}^{Z-1}$, $\beta > 1$, $Z \in \{2, 3\}$, choosing

$$\delta' = \arg\max_{\eta \in \mathcal{S}} \ell\Big(f_\theta\Big(x + \Pi_\Delta(\hat{\delta}(x) + \eta v)\Big), y\Big). \quad (12)$$

Since $\alpha \in \mathcal{S}$ and each candidate is projected onto $\Delta$, LS-PGD is at least as strong as a standard single PGD step from the same warm start, while often recovering multi-step progress using only a few forward probes.

**Curriculum score and scheduling.** From Eqn. 7, samples with small $\underline{m}(x; \theta)$ incur large robust-hinge values and dominate robust risk. We therefore define a perturbation score as the robust-hinge *evaluated at the estimated margin*:

$$s(x) := \big[1 - \widehat{m}_{\mathrm{rob}}(x; \theta)\big]_+, \quad (13)$$

so that $s(x)$ is larger for smaller estimated robust margins. At each epoch, we compute $s(x)$ for adversarial samples, sort them in descending order, and construct mini-batches following this order. For each mini-batch, we optimize the robustness term (Eqn. 5) using its adversarial companions and the current synthetic set. This schedule implements the theoretical prescription of Sec. 4.1: it concentrates updates on the low-margin tail, thereby directly targeting an increase of the minimum robust margin induced by the distilled set.

### 4.3. Contrastive Robustness Loss

The robust-margin formulation (Eqn. 6) makes explicit that robustness is determined by the closest competing class

through the term $\max_{k \neq y} f_k(x + \delta)$. In particular, a sample becomes non-robust when some impostor logit overtakes the true logit under an admissible perturbation. This observation exposes a mismatch in class-mean robustness alignment (Eqn. 5): matching $\mathbb{E}[e(x)]$ and $\mathbb{E}[e(\tilde{x})]$ within each class can increase average invariance but provides no explicit pressure to separate an example from its most competitive negatives, i.e., those nearest to the decision boundary. Consistent with Sec. 4.1 and AAC (Sec. 4.2), we therefore replace mean matching with an instance-level objective that (i) enforces clean–adversarial invariance for hard (low-margin) cases, and (ii) simultaneously repels nearest impostors.

**Instance-level supervised contrast for robustness.** Given a mini-batch $B = \{(x_i, y_i)\}_{i=1}^M$ and corresponding adversarial companions $\tilde{x}_i$ (generated and ordered by AAC), let $e(\cdot) \in \mathbb{R}^d$ be the embedding and $sim(u, v)$ a cosine similarity. For each anchor $x_i$, we define the *positive set*

$$P(i) := \{\tilde{x}_i\} \cup \{x_j, \tilde{x}_j \mid y_j = y_i, \ j \neq i\}, \qquad (14)$$

and the candidate set $A(i)$ as the union of positives and all remaining (different-class) examples:

$$A(i) := P(i) \cup \{x_k, \tilde{x}_k \mid y_k \neq y_i\}. \qquad (15)$$

Define logits $g_{i,a} := sim(e(x_i), e(a))$ and a temperature $\tau > 0$. We use a supervised contrastive objective as the *Contrastive Robustness Loss (CRL)*:

$$\mathcal{L}_{\text{CRL}} = \frac{1}{M} \sum_{i=1}^M \left[ -\sum_{a \in P(i)} \frac{1}{|P(i)|} \log \frac{\exp(g_{i,a}/\tau)}{\sum_{b \in A(i)} \exp(g_{i,b}/\tau)} \right]. \qquad (16)$$

CRL matches the margin view: increasing the numerator enhances clean–adversarial invariance (raising the true-class score under perturbations), while the normalization term penalizes high similarity to all competing classes, with the largest effect on the hardest negatives (those with largest $g_{i,b}$). Thus, minimizing $\mathcal{L}_{\text{CRL}}$ (i) tightens the clean–adversarial neighborhood within class and (ii) suppresses the closest impostors that control $\max_{k \neq y} f_k(\cdot)$ in Eqn. 6.

**Coupling with the attack-aware curriculum.** AAC ranks adversarial samples by $s(x)$ (Eqn. 13), concentrating optimization on low-margin cases. Applying CRL on these ordered batches targets the tail events that dominate robust risk (Sec. 4.1): anchors with small $\widehat{m}_{\text{rob}}$ are forced to (a) stay close to their adversarial counterparts and (b) move away from their nearest competing-class embeddings, thereby pushing up the minimum robust margin of the classifier.

**Efficient class-balanced memory queue.** A full denominator over all in-batch pairs scales as $\mathcal{O}(M^2)$ and limits the number of negatives. To obtain many informative negatives at modest cost, we maintain a *class-balanced* memory queue. For each class $c$, a FIFO queue stores up to $Q$ cached

embeddings with labels. For an anchor $x_i$ of class $y_i$, we retrieve hard negatives by approximate search: we form a low-dimensional proxy $\tilde{e}(x) = R\,e(x) \in \mathbb{R}^r$ using a fixed random projection $R \in \mathbb{R}^{r \times d}$ with $r \ll d$, compute similarities between $\tilde{e}(x_i)$ and proxies in queues of classes $c \neq y_i$, and select the top-$k$ most similar items as hard negatives. We then evaluate Eqn. 16 using the *full* embeddings for these retrieved negatives (the projection is used only for retrieval). This reduces the per-step contrastive computation from $\mathcal{O}(M^2)$ to $\mathcal{O}(Mk)$ while preserving the key pressure to repel the closest impostors that dominate robustness.

### 4.4. Optimization of C²R

Our method directly follows the margin view in Sec. 4.1: robustness is driven by low-margin samples. We therefore (i) use AAC to emphasize hard adversarial cases (Sec. 4.2), and (ii) replace class-mean robustness alignment with the instance-level CRL (Sec. 4.3).

**Objective.** We keep the standard performance term and use CRL as the robustness term:

$$\mathcal{L}_{\text{C}^2\text{R}} = (1 - \eta)\mathcal{L}_{\text{perf}} + \eta\,\mathcal{L}_{\text{CRL}}, \qquad \eta \in [0, 1]. \qquad (17)$$

**How it is optimized.** AAC adds no extra loss. Each epoch, we generate adversarial companions with LS-PGD, compute the score $s(x)$ (Eqn. 13), and form mini-batches by sorting samples from hard to easy. We then optimize Eqn. 17 on these curriculum-ordered batches, so CRL mainly acts on the low-margin region that controls robust risk.

## 5. Experiment

### 5.1. Experimental Setup

**Datasets and Networks.** We evaluate our method on several standard DD benchmarks: CIFAR-10 and CIFAR-100 (Krizhevsky et al., 2009) and Tiny-ImageNet (Le & Yang, 2015). For ImageNet-1K (Russakovsky et al., 2015), we adopt six subsets (ImageNette, ImageWoof, ImageFruit, ImageMeow, ImageSquawk, and ImageYellow), resizing all images to $128 \times 128$ to balance resolution and cost. Following DD-RobustBench and prior DD work (Zhao et al., 2021; Zhao & Bilen, 2023; Wu et al., 2025), we employ the standard three-block ConvNet (Gidaris & Komodakis, 2018) on CIFAR-10/100, and use ConvNet-D4 and ConvNet-D5 on higher-resolution datasets. More experiments of datasets and networks are provided in Appendix C.

**Baselines.** We compare our C²R to only robustness-oriented DD baselines that provide official implementations and compatible benchmark settings for fair comparison. We include MTT (Cazenavette et al., 2022), SRe²L (Yin et al., 2023), D⁴M (Su et al., 2024), and ROME (Zhou et al., 2025). More details of baselines are provided in Appendix B.3.

*Table 1.* **Test accuracy (%) of baseline methods and C²R** on CIFAR-10, CIFAR-100, and Tiny-ImageNet with IPC = 1, 5, 10, 30, 50, evaluated under adversarial attacks with $|\varepsilon| = 2/255$. Relative improvements over the second-best method in each setting are shown as subscripts highlighted in blue. All results are averaged over five independent runs.

| Methods | Attack | CIFAR-10 | | | | | CIFAR-100 | | | | | Tiny-ImageNet | | | | |
|---|---|---|---|---|---|---|---|---|---|---|---|---|---|---|---|---|
| | | 1 | 5 | 10 | 30 | 50 | 1 | 5 | 10 | 30 | 50 | 1 | 5 | 10 | 30 | 50 |
| SRe²L | Clean | 13.49 | 31.06 | 37.53 | 54.88 | 63.28 | 4.68 | 28.92 | 39.64 | 51.33 | 53.95 | 6.28 | 18.38 | 26.92 | 39.49 | 43.24 |
| | FGSM | 7.67 | 15.09 | 14.94 | 18.89 | 21.39 | 2.39 | 8.47 | 10.96 | 14.17 | 15.08 | 1.73 | 1.75 | 2.69 | 4.77 | 5.71 |
| | PGD | 7.17 | 13.95 | 13.09 | 15.29 | 16.12 | 2.35 | 6.85 | 7.08 | 7.69 | 8.04 | 1.50 | 1.11 | 1.59 | 2.40 | 2.70 |
| | CW | 6.17 | 13.77 | 12.92 | 15.53 | 16.76 | 1.62 | 4.46 | 4.97 | 6.26 | 6.27 | 1.26 | 1.03 | 1.67 | 2.70 | 2.94 |
| | VMI | 6.96 | 14.09 | 13.28 | 15.39 | 15.73 | 2.34 | 5.72 | 5.48 | 5.62 | 5.49 | 1.52 | 1.11 | 1.54 | 2.40 | 2.60 |
| | Jitter | 6.22 | 14.40 | 14.07 | 17.38 | 19.49 | 1.56 | 4.66 | 5.93 | 5.93 | 8.29 | 1.23 | 0.88 | 1.43 | 2.35 | 2.72 |
| D⁴M | Clean | 23.39 | 42.34 | 48.16 | 63.53 | 69.81 | 11.27 | 31.90 | 40.12 | 48.15 | 51.10 | 2.16 | 6.94 | 14.25 | 35.80 | 42.89 |
| | FGSM | 9.67 | 17.19 | 22.07 | 25.72 | 27.13 | 3.37 | 5.91 | 6.33 | 6.40 | 5.17 | 0.63 | 1.37 | 1.81 | 3.92 | 6.30 |
| | PGD | 8.86 | 15.47 | 20.14 | 22.20 | 23.08 | 2.84 | 4.18 | 4.25 | 3.97 | 3.14 | 0.53 | 1.05 | 0.97 | 1.91 | 3.14 |
| | CW | 8.35 | 15.38 | 20.16 | 22.62 | 23.71 | 2.84 | 4.61 | 4.68 | 4.28 | 3.46 | 0.50 | 0.98 | 1.12 | 2.29 | 3.51 |
| | VMI | 8.92 | 15.50 | 20.14 | 22.33 | 23.22 | 2.86 | 4.21 | 4.14 | 3.79 | 2.92 | 0.52 | 1.04 | 0.96 | 1.70 | 2.85 |
| | Jitter | 8.24 | 15.20 | 19.85 | 22.95 | 24.58 | 3.36 | 5.17 | 5.19 | 5.00 | 3.80 | 0.48 | 0.90 | 1.06 | 1.97 | 3.10 |
| ROME | Clean | 33.25 | 39.78 | 47.94 | 54.08 | 60.21 | 13.76 | 23.08 | 34.73 | 40.57 | 46.41 | 3.36 | 8.49 | 14.90 | 20.36 | 25.82 |
| | FGSM | 21.32 | 23.28 | 25.72 | 25.42 | 25.12 | 7.18 | 8.20 | 9.48 | 9.96 | 10.43 | 1.36 | 1.48 | 1.64 | 2.08 | 2.52 |
| | PGD | 20.21 | 21.90 | 24.01 | 22.45 | 20.90 | 5.63 | 6.87 | 8.42 | 7.83 | 7.23 | 1.02 | 1.17 | 1.36 | 1.59 | 1.83 |
| | CW | 19.84 | 22.23 | 25.22 | 24.17 | 23.12 | 3.90 | 6.33 | 9.37 | 8.09 | 6.81 | 1.39 | 1.34 | 1.28 | 1.43 | 1.57 |
| | VMI | 20.97 | 22.37 | 24.12 | 23.88 | 23.64 | 5.14 | 7.34 | 10.10 | 8.47 | 6.84 | 1.86 | 1.69 | 1.47 | 1.35 | 1.23 |
| | Jitter | 21.66 | 23.14 | 24.98 | 24.88 | 24.78 | 5.51 | 7.36 | 9.67 | 8.37 | 7.07 | 1.27 | 1.35 | 1.44 | 1.54 | 1.64 |
| C²R | Clean | 35.04 | 40.94 | 48.31 | 56.52 | 64.73 | 14.33 | 24.46 | 37.12 | 42.69 | 48.25 | 5.38 | 10.99 | 18.00 | 22.77 | 27.53 |
| | FGSM | 23.64(+2.32) | 25.68(+2.40) | 28.24(+2.52) | 27.88(+2.16) | 27.53(+0.40) | 8.94(+1.76) | 10.58(+2.11) | 12.63(+1.67) | 15.88(+1.71) | 15.72(+0.64) | 3.13(+1.40) | 3.79(+2.04) | 4.61(+1.92) | 5.78(+1.01) | 6.95(+0.65) |
| | PGD | 23.84(+3.63) | 26.19(+4.29) | 29.12(+5.11) | 28.08(+5.63) | 27.04(+3.96) | 8.11(+2.48) | 9.56(+2.69) | 11.38(+2.96) | 12.00(+4.17) | 12.61(+4.57) | 2.05(+0.55) | 2.84(+1.67) | 3.83(+2.24) | 4.38(+1.98) | 4.92(+1.78) |
| | CW | 24.07(+4.23) | 26.24(+4.01) | 28.95(+3.73) | 28.10(+3.93) | 27.25(+3.54) | 7.23(+3.33) | 9.67(+3.34) | 12.72(+3.35) | 12.04(+3.95) | 11.35(+4.54) | 2.02(+0.63) | 2.57(+1.23) | 3.25(+1.58) | 4.04(+1.34) | 4.83(+1.32) |
| | VMI | 23.93(+2.96) | 25.96(+3.59) | 28.49(+4.37) | 27.71(+3.83) | 26.94(+3.30) | 7.29(+2.15) | 9.79(+2.45) | 12.92(+2.82) | 11.77(+3.30) | 10.62(+3.78) | 2.48(+0.62) | 2.83(+1.14) | 3.27(+1.73) | 3.69(+1.29) | 4.12(+1.27) |
| | Jitter | 24.83(+3.17) | 25.92(+2.78) | 27.29(+2.31) | 27.34(+2.46) | 27.21(+2.43) | 7.58(+2.07) | 9.49(+2.13) | 11.87(+2.20) | 11.73(+3.36) | 11.59(+3.30) | 2.35(+1.08) | 2.97(+1.62) | 3.75(+2.31) | 4.38(+2.03) | 5.01(+1.91) |

*Table 2.* **Test accuracies (%) of baseline methods and C²R** with IPC = 1, 10, 50 on ImageNet-1K subsets, evaluated under adversarial attacks with $|\varepsilon| = 2/255$. All results are averaged over five independent runs.

| Method | | ImageNette | | | ImageWoof | | | ImageFruit | | | ImageMeow | | | ImageSquawk | | | ImageYellow | | |
|---|---|---|---|---|---|---|---|---|---|---|---|---|---|---|---|---|---|---|---|---|
| | | 1 | 10 | 50 | 1 | 10 | 50 | 1 | 10 | 50 | 1 | 10 | 50 | 1 | 10 | 50 | 1 | 10 | 50 |
| MTT | Clean | 48.20 | 66.40 | 67.60 | 30.40 | 38.00 | 39.40 | 25.00 | 42.20 | 44.60 | 31.00 | 44.40 | 44.20 | 39.00 | 55.60 | 59.20 | 44.60 | 63.40 | 66.20 |
| | FGSM | 19.40 | 27.80 | 25.20 | 6.20 | 5.80 | 3.40 | 6.80 | 11.20 | 9.80 | 5.40 | 8.00 | 4.00 | 11.80 | 16.20 | 13.60 | 17.20 | 23.40 | 21.20 |
| | PGD | 18.60 | 23.60 | 20.60 | 5.20 | 4.00 | 1.80 | 6.00 | 9.20 | 7.20 | 5.00 | 5.60 | 2.80 | 10.60 | 12.40 | 11.00 | 15.80 | 18.80 | 17.40 |
| | CW | 17.60 | 22.60 | 20.60 | 3.60 | 3.40 | 1.60 | 5.40 | 9.00 | 7.60 | 4.40 | 5.20 | 3.00 | 10.20 | 12.00 | 10.60 | 14.40 | 18.20 | 17.20 |
| | VMI | 18.60 | 23.80 | 19.80 | 5.60 | 4.20 | 1.80 | 6.00 | 9.00 | 7.20 | 5.00 | 5.60 | 2.80 | 10.80 | 12.40 | 11.60 | 15.80 | 19.00 | 17.40 |
| | Jitter | 17.80 | 23.80 | 23.40 | 4.20 | 4.60 | 2.20 | 6.00 | 10.00 | 9.00 | 4.60 | 6.40 | 4.60 | 11.00 | 14.00 | 13.60 | 15.40 | 21.00 | 20.20 |
| C²R | Clean | 44.10 | 63.80 | 64.70 | 28.50 | 36.30 | 37.40 | 22.80 | 39.60 | 44.20 | 27.50 | 44.20 | 44.60 | 35.00 | 54.20 | 58.90 | 42.70 | 60.30 | 63.80 |
| | FGSM | 22.50 | 30.30 | 27.90 | 7.40 | 8.20 | 6.80 | 7.80 | 14.40 | 14.00 | 7.50 | 10.60 | 7.10 | 13.60 | 18.90 | 16.80 | 18.10 | 25.90 | 25.30 |
| | PGD | 19.90 | 26.10 | 23.80 | 8.30 | 7.40 | 5.10 | 7.40 | 13.80 | 10.50 | 6.20 | 10.00 | 5.70 | 14.20 | 15.20 | 14.80 | 16.90 | 20.20 | 20.00 |
| | CW | 20.00 | 24.10 | 24.10 | 5.10 | 5.00 | 2.70 | 6.80 | 11.90 | 11.20 | 5.90 | 8.60 | 5.90 | 12.10 | 14.80 | 14.20 | 16.10 | 21.10 | 20.40 |
| | VMI | 20.10 | 26.90 | 21.20 | 5.70 | 5.70 | 2.60 | 8.20 | 10.20 | 9.90 | 6.20 | 7.10 | 3.10 | 12.40 | 14.80 | 13.20 | 16.50 | 22.10 | 19.00 |
| | Jitter | 19.20 | 25.70 | 25.60 | 5.60 | 7.10 | 4.10 | 7.90 | 13.20 | 11.20 | 7.00 | 9.10 | 7.10 | 13.10 | 17.70 | 17.20 | 17.10 | 24.30 | 23.10 |

**Evaluation Attacks.** Following the adversarial evaluation protocol in DD-RobustBench (Wu et al., 2025), we evaluate robustness under six widely used attacks: FGSM (Goodfellow et al., 2015), PGD (Madry et al., 2018), CW (Carlini & Wagner, 2017), VMIFGSM (Wang & He, 2021), Jitter (Schwinn et al., 2023), and AutoAttack (Liu et al., 2022). For all methods except AutoAttack, we use an $\ell_\infty$ perturbation budget of $|\varepsilon| = 2/255$ in the main text, while the details of AutoAttack are provided in Appendix C.1.

**Training Settings.** In the distillation stage, we consider images-per-class (IPC) budgets of $1, 10, 50$. All models are optimized with SGD using a learning rate of $0.01$, momentum of $0.9$, and weight decay of $5 \times 10^{-4}$. Additional implementaion details are provided in Appendix B.

## 5.2. Comparisons with the State-of-the-art

**Main Results.** We comprehensively evaluate the clean and robust performance of C²R compared with representative robust DD methods. As summarized in Tab. 1 and Tab. 2, C²R consistently achieves the highest accuracy across all datasets and attack types ($|\varepsilon| = 2/255$), notably surpassing other methods by an average of 2.8%. The improvement is most pronounced at high IPC values, where the larger synthetic data budget allows the AAC to refine robustness over a denser sample distribution, and the CRL more effectively enlarges the robust margin by aligning features. This synergy yields a better balance of accuracy and robustness. More experiments are provided in Appendix C.1.

In Tab. 1 and Tab. 2, we further observe that C²R exhibits stable gains across diverse attacks rather than favoring a

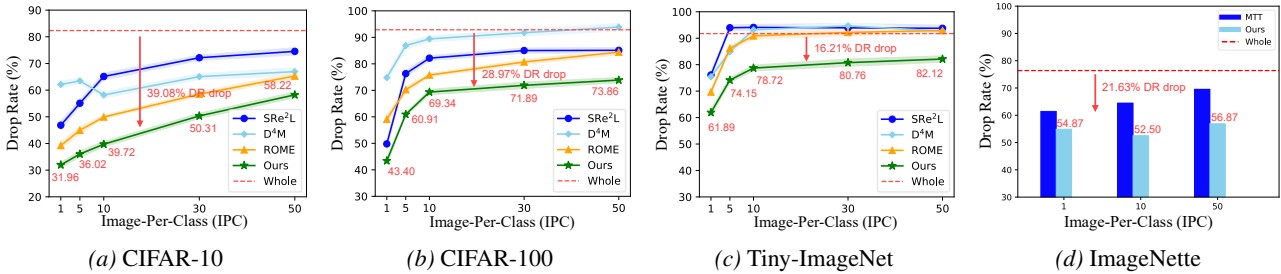

*Figure 3.* **Drop rate (DR) comparison under PGD attacks across datasets and IPC.** The red dashed line denotes the DR of models trained on the whole dataset. C$^2$R consistently achieves the lowest drop rate and exhibits a flatter trajectory as the synthetic set scales.

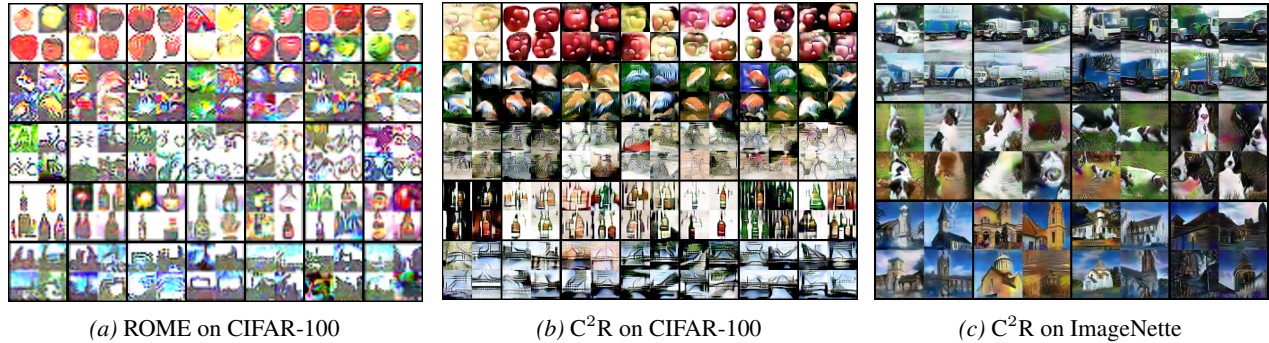

*(a)* ROME on CIFAR-100     *(b)* C$^2$R on CIFAR-100     *(c)* C$^2$R on ImageNette

*Figure 4.* **Visualization of synthetic images** distilled by ROME on CIFAR-100, and C$^2$R on CIFAR-100 and ImageNette under IPC = 50.

specific threat model, indicating that the robustness improvements stem from distribution-level alignment rather than attack-specific adaptation. However, we also observe a slight decrease in robust accuracy when IPC increases from 10 to 50, which is because denser synthetic distributions lead to more overlapping local regions near decision boundaries, where adversarial perturbations can more easily transfer across neighboring samples.

**Robustness Degradation under Adversarial Perturbations.** We further examine how different DD methods respond to adversarial perturbations using the drop rate (DR) metric from DD-RobustBench (Wu et al., 2025), defined as $(\text{Acc}_{\text{original}} - \text{Acc}_{\text{robust}})/\text{Acc}_{\text{original}} \times 100\%$. As shown in Fig. 3, C$^2$R consistently achieves the lowest DR across all datasets and IPC settings, with its curve remaining below those of competing methods, indicating the smallest accuracy degradation. Under PGD evaluation, the average DR of C$^2$R across datasets stays below 66.8%, to our knowledge the first DD method to reach this regime. In addition, while most baselines exhibit rapidly increasing DR as IPC grows, C$^2$R maintains a much flatter trajectory, suggesting that the robustness of C$^2$R remains more stable as the synthetic dataset scales. Additional evaluations under FGSM are provided in Appendix C.2.

**Visualization Results.** We qualitatively assess the distilled datasets by comparing synthetic samples from C$^2$R and ROME on CIFAR-100 and ImageNette with IPC = 50.

As shown in Fig. 4, C$^2$R generates images with clearer object structures, more coherent color composition, and more recognizable semantics, whereas ROME often exhibits color artifacts and repetitive patterns. The AAC encourages the generator to emphasize boundary-sensitive instances, leading to more discriminative patterns, while CRL enforces local feature consistency and suppresses redundant textures.

**Training Time Comparison.** We assess efficiency by plotting average accuracy against cost in Fig. 5. C$^2$R consistently achieves higher accuracy with noticeably smaller training time than SRe$^2$L and ROME, indicating stronger robustness under lower cost. This gap stems from two reasons: AAC cuts redundant adversarial updates via LS-PGD, while CRL stabilizes optimization and speeds convergence. More experiments are provided in Appendix C.

### 5.3. Ablation Study

**Components of Backbone.** We ablate the AAC and CRL on benchmarks under identical backbones and optimization settings, toggling each component as in Tab. 3. Removing both modules leads to a clear drop in robust accuracy, while enabling either AAC or CRL alone yields noticeable gains over the baseline. Activating both simultaneously produces the largest improvements, consistently outperforming either single-module variant. This suggests that AAC and CRL are complementary: AAC prioritizes smallest-margin adversarial examples, and CRL exploits this ordered stream to

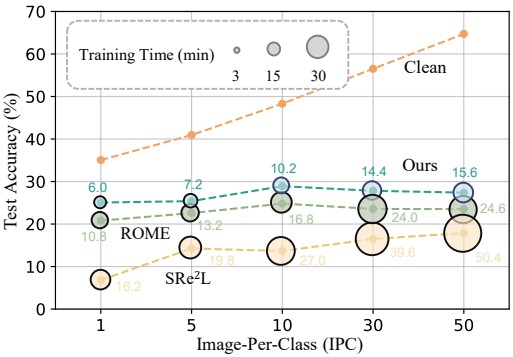

*Figure 5.* **Validation accuracy vs. distillation time on CIFAR-**10. The orange dashed curve reports clean accuracy.

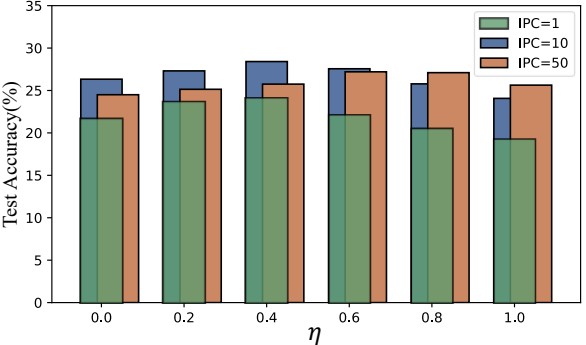

*Figure 6.* **Ablation study of the $\eta$ on CIFAR-10.** The results denote the average robust accuracy across all attack methods.

*Table 3.* **Ablation on $C^2R$ components.** Average robust accuracy over all attacks on CIFAR-10, CIFAR-100, Tiny-ImageNet, and ImageNette with IPC = 10. Improvements are highlighted in blue.

| AAC | CRL | CIFAR-10 | CIFAR-100 | Tiny-ImageNet | ImageNette |
|---|---|---|---|---|---|
| ✗ | ✗ | 24.81 | 9.40 | 1.43 | 23.76 |
| ✓ | ✗ | 26.73(+1.92) | 11.45(+2.05) | 2.73(+1.30) | 25.27(+1.51) |
| ✗ | ✓ | 26.04(+1.23) | 10.98(+1.58) | 2.10(+0.67) | 24.65(+0.89) |
| ✓ | ✓ | **28.41**(+3.60) | **12.30**(+2.90) | **3.74**(+2.31) | **26.62**(+2.86) |

*Table 4.* **Average robust accuracy and efficiency of $C^2R$ under two inner attackers** on four datasets when IPC=10. Accuracy is averaged, and speedup is relative to the multi-step PGD attacker.

| Attacker | CIFAR-10 | CIFAR-100 | Tiny-Imagenet | Imagenette | Speedup |
|---|---|---|---|---|---|
| PGD | **28.63** | **12.54** | **3.92** | **26.82** | 1× |
| **LS-PGD (ours)** | 28.41 | 12.30 | 3.74 | 26.62 | **1.4×** |

enforce clean–adversarial invariance and stronger inter-class separation near decision boundaries, yielding a robustness level unattainable by either component in isolation.

**Impact of the Hyperparameter $\eta$.** We study the sensitivity of $C^2R$ to the weighting factor $\eta$ in Eqn. 17 by varying $\eta$ under different IPC settings. As shown in Fig. 6, the optimal $\eta$ depends on the synthetic budget: $\eta = 0.4$ yields the best performance when IPC is 1 or 10, while $\eta = 0.6$ becomes preferable when IPC increases to 50. Across all settings, very small $\eta$ underutilizes the contrastive robustness signal and leads to weaker robust margins, whereas overly large $\eta$ starts to over-regularize the model, slightly hurting accuracy as the training is dominated by adversarial invariance.

These trends are consistent with the robust-margin perspective of $C^2R$. Under low IPC, the synthetic dataset is sparse and the contrastive signal is inherently noisy; a smaller $\eta$ helps maintain task alignment by preventing the contrastive loss from overfitting to a few hard pairs. As IPC grows, the synthetic set becomes more diverse, providing enough samples to support reliable contrastive alignment. In this context, a larger $\eta$ allows the contrastive robustness loss to play a more dominant role. Overall, we observe a broad

plateau around the best $\eta$ values rather than a sharp optimum, indicating that $C^2R$ is not sensitive to moderate misspecification of $\eta$. Based on these observations, we adopt $\eta = 0.4$ for IPC $= 1$ and 10, and $\eta = 0.6$ for IPC $= 50$ as default settings, striking a balance between performance alignment and contrastive robustness.

**Impact of LS-PGD.** To evaluate the influence of the LS-PGD, we compare $C^2R$ under two configurations on CIFAR-10 with IPC$= 10$. The first configuration uses a standard multi-step PGD, while the second one corresponds to the $C^2R$ equipped with the proposed LS-PGD. As shown in Tab. 4, the LS-PGD achieves accuracy comparable to or slightly better than the multi-step PGD while using substantially less time. LS-PGD preserves the strength of multi-step PGD, enabling reliable ordered adversaries at a fraction of the cost. Overall, this ablation confirms that LS-PGD delivers a favorable robustness–efficiency trade-off. Additionally, this also indicates that $C^2R$ is not sensitive to the specific PGD solver quality as long as the adversary is sufficiently strong to induce a meaningful margin ordering.

## 6. Conclusion

In this work, we revisited DD from the perspective of adversarial robustness, asking how to endow compact synthetic datasets with reliable performance under strong attacks. We introduced **$C^2R$**, a Contrastive Curriculum for Robust Dataset Distillation that couples an *Attack-Aware Curriculum* (AAC), which orders samples by robust margin via a perturbation score, with a *Contrastive Robustness Loss* (CRL) that aligns clean–adversarial features while separating classes near the decision boundary. Together with an efficient LS-PGD attacker, $C^2R$ consistently improves robust accuracy while maintaining clean accuracy, achieving around an average of 2.8% higher robust accuracy than robust DD baselines, lower robustness degradation as IPC grows, and favorable training-time and memory costs. Limitations and future work are provided in Appendix D.

## Impact Statement

This work advances dataset distillation by improving the accuracy-robustness trade-off under adversarial evaluations, which benefits applications that require reliable learning under limited storage/compute and robustness constraints. We clearly scope our claims to the stated perturbation budgets and attack protocols, recommend reporting robustness under multiple strong attacks, and encourage practitioners to conduct privacy audits and follow applicable data governance policies in real-world systems. Overall, we do not anticipate additional societal consequences beyond those commonly associated with developing more efficient and robust machine learning methods.

## Acknowledgments

This research was partially supported by the National Natural Science Foundation of China (NSFC) (granted No. 62306064) and the Fundamental Research Funds (granted No. ZYGX2025XJ049). We appreciate all the authors for their fruitful discussions. In addition, thanks are extended to anonymous reviewers for their insightful comments.

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

# A. Algorithm

---

**Algorithm 1** Pseudocode of Contrastive Curriculum for Robust Dataset Distillation (C$^2$R)

---

1: **Input:** real dataset $\mathcal{D} = \{(x_i, y_i)\}$; IPC budget $N$; student network $f_\theta$; trade-off weight $\eta$; perturbation budget $\varepsilon$; attack steps $T$; distillation iterations $I$; memory size $Q$.
2: **Output:** synthetic dataset $X = \{(x_s, y_s)\}_{s=1}^N$.
3: Initialize synthetic images and labels $X$.
4: Initialize LS-PGD warm-starts $\hat{\delta}(x) \leftarrow 0$ for $x \in \mathcal{D}$.
5: Initialize memory queues $\{\mathcal{Q}_c\}_{c=1}^C$ as empty (capacity $Q$).
6: **for** $i = 1$ **to** $I$ **do**
7:     Sample a mini-batch $B = \{(x_r, y_r)\}_{r=1}^M$ from $\mathcal{D}$.
8:     **for** each $(x_r, y_r) \in B$ **do**
9:         Use LS-PGD to obtain adversarial image $\tilde{x}_r$.
10:         Estimate robust-margin surrogate $\widehat{m}_{\text{rob}}(x_r; \theta)$ and score $s(x_r)$.
11:         Update warm-start $\hat{\delta}(x_r) \leftarrow \tilde{x}_r - x_r$.
12:     **end for**
13:     Sort $\{(x_r, \tilde{x}_r, y_r)\}_{r=1}^M$ by descending $s(x_r)$ and split into curriculum mini-batches $\{\mathcal{B}_t\}$.
14:     **for** each curriculum mini-batch $\mathcal{B}_t$ **do**
15:         Sample a synthetic batch $B_t^s \subset X$ with labels matching $\mathcal{B}_t$.
16:         Generate adversarial counterparts $\widetilde{B_t^s}$ for $B_t^s$ using LS-PGD.
17:         Compute $\mathcal{L}_{\text{perf}}$ on $B_t^s$ via cross-entropy.
18:         Build positives $P(i)$ and negatives $N(i)$ using: its adversary in $\widetilde{B_t^s}$, same-class samples (and adversaries), and hard negatives retrieved from $\{\mathcal{Q}_c\}$.
19:         Compute contrastive robustness loss $\mathcal{L}_{\text{CRL}}$.
20:         Form total loss $\mathcal{L}_{\text{C}^2\text{R}} \leftarrow (1 - \eta)\mathcal{L}_{\text{perf}} + \eta\mathcal{L}_{\text{CRL}}$.
21:         Update $\theta$ and $X$ by one SGD step on $\mathcal{L}_{\text{C}^2\text{R}}$.
22:         Enqueue embeddings of samples in $B_t^s$ (and optionally $\widetilde{B_t^s}$) into the corresponding queues, and discard oldest to keep $|\mathcal{Q}_c| \leq Q$.
23:     **end for**
24: **end for**
25: **return** $\mathcal{X}$.

---

# B. More Experimental Settings

## B.1. Datasets

We conduct experiments on several widely used vision datasets covering low-, mid-, and high-resolution settings.

- **CIFAR-10** (Krizhevsky et al., 2009). The dataset contains 50K training images and 10K test images at $32 \times 32$ across 10 classes.

- **CIFAR-100** (Krizhevsky et al., 2009). The dataset follows the same image format as CIFAR-10 but extends the label set to 100 categories, with 600 images provided for each class.

- **Tiny-ImageNet** (Le & Yang, 2015). The dataset provides 200 categories, each with 500 images at a resolution of $64 \times 64$.

- **ImageNet-1K Subsets** (Russakovsky et al., 2015). These datasets group ImageNet classes into several 10-class subsets following prior practice, and all images are standardized at $128 \times 128$ resolution using the official split. The ImageNet subsets used in our experiments cover several 10-class groups, including ImageNette, ImageWoof, ImageFruit, ImageMeow, ImageSquawk, and ImageYellow. In addition, we include ImageNet-10 (Kim et al., 2022), constructed by selecting the first ten classes from the ImageNet-1K class index list.

## B.2. Adversarial Attack Methods

We employ eight adversarial attacks to evaluate robustness.

- **Fast Gradient Sign Method (FGSM)** (Goodfellow et al., 2015). The attack perturbs the input by a single step in the direction of the sign of the loss gradient under an $\ell_\infty$ bound.

- **Projected Gradient Descent (PGD)** (Madry et al., 2018). The attack iteratively applies small step updates of the loss gradient and projects the perturbed sample back into the $\ell_\infty$ ball.

- **Carlini & Wagner (CW)** (Carlini & Wagner, 2017). The attack formulates adversarial example generation as a constrained problem, replacing the hard classification constraint with a surrogate loss and minimizing the $\ell_p$-bounded perturbation.

- **Variance-Tuned Momentum Iterative FGSM (VMIFGSM)** (Wang & He, 2021). The attack improves iterative FGSM by adding momentum and tuning the gradient direction using variance estimated from sampled neighbors, which stabilizes updates and enhances transferability.

- **Jitter** (Schwinn et al., 2023). The attack modifies iterative gradient-based updates by injecting Gaussian noise into the model logits and rescaling them, which encourages exploration of diverse gradient directions.

- **AutoAttack** (Liu et al., 2022). The attack integrates several strong and complementary adversarial methods, executed with fixed hyperparameter settings, forming a deterministic and parameter-free evaluation pipeline.

- **Square** (Andriushchenko et al., 2020). The attack performs black-box optimization through local search using square-shaped, updating the adversarial example only when the sampled modification reduces the loss.

- **Momentum Iterative Method (MIM)** (Dong et al., 2018). The attack augments iterative FGSM by accumulating a momentum term across iterations to stabilize update directions and avoid poor local maxima, thereby improving both white-box strength and black-box transferability.

## B.3. More Details on Baselines

- **MTT** (Cazenavette et al., 2022) learns a small synthetic dataset such that training a model on it produces parameter trajectories that closely match those obtained when training on the full real dataset.

- **SRe$^2$L** (Yin et al., 2023) learns ImageNet-scale distilled datasets through a three-stage squeeze–recover–relabel framework, where synthetic images are recovered by matching BatchNorm statistics with multi-crop optimization and then refined with soft labels from the teacher model.

- **D$^4$M** (Su et al., 2024) constructs distilled datasets by generating class-representative images from disentangled diffusion models using latent space prototypes and text conditions.

- **ROME** (Zhou et al., 2025) formulates robust dataset distillation under an information bottleneck framework, where synthetic data are optimized through performance alignment with a robust model and class-wise robustness alignment to adversarial priors in the feature embedding space.

- **GUARD** (Xue et al., 2025) enhances adversarial robustness in dataset distillation through curvature-based geometric regularization, where the loss curvature is controlled by penalizing gradient variations along normalized ascent directions.

- **RDD** (Vahidian et al., 2025) adopts a Double-DRO distillation scheme that clusters real samples and minimizes CVaR-based losses during model updates, producing robust synthetic data.

# C. More Experimental Results

## C.1. More Comparisons with Previous Methods

To complement the analyses in the main paper, we provide additional robustness results for baseline methods and C$^2$R under multiple perturbation budgets, AutoAttack, and ImageNet-1K subsets. The complete results are reported in Tab. 5-Tab. 9. Across CIFAR-10, CIFAR-100, Tiny-ImageNet, and ImageNet-1K subsets, C$^2$R consistently achieves higher robust accuracy than the baselines in most settings, and the gains remain stable under both $|\varepsilon| = 4/255$ and $|\varepsilon| = 8/255$. For AutoAttack, C$^2$R maintains the highest accuracy under most IPC level and dataset, showing that its margin-oriented design continues to control the hardest adversarial cases. We observe that the combination of Attack-Aware Curriculum and Contrastive Robustness Loss enlarges the smallest robust margin even when perturbation strength increases or when the dataset exhibits higher semantic diversity. These results confirm that C$^2$R generalizes beyond the main experimental setup and preserves its robustness advantage across perturbation budgets and IPC.

*Table 5.* **Test accuracy (%) of baseline methods and $C^2R$** on CIFAR-10, CIFAR-100, and Tiny-ImageNet with IPC = 1, 5, 10, 30, 50, evaluated under adversarial attacks with $|\varepsilon| = 4/255$. Relative improvements over the second-best method in each setting are shown as subscripts highlighted in blue. All results are averaged over five independent runs.

| Methods | Attack | CIFAR-10 | | | | | CIFAR-100 | | | | | Tiny-ImageNet | | | | |
|---|---|---|---|---|---|---|---|---|---|---|---|---|---|---|---|---|
| | | 1 | 5 | 10 | 30 | 50 | 1 | 5 | 10 | 30 | 50 | 1 | 5 | 10 | 30 | 50 |
| $D^4M$ | Clean | 23.39 | 42.34 | 48.16 | 63.53 | 69.81 | 11.27 | 31.90 | 40.12 | 48.15 | 51.10 | 2.16 | 6.94 | 14.25 | 35.80 | 42.89 |
| | FGSM | 3.62 | 6.19 | 9.05 | 8.26 | 8.50 | 1.25 | 1.15 | 1.07 | 1.12 | 0.87 | 0.29 | 0.50 | 0.53 | 1.18 | 1.83 |
| | PGD | 2.44 | 3.62 | 5.86 | 4.02 | 3.84 | 0.78 | 0.35 | 0.36 | 0.29 | 0.13 | 0.23 | 0.22 | 0.16 | 0.13 | 0.30 |
| | CW | 2.44 | 3.94 | 6.48 | 4.46 | 4.26 | 0.85 | 0.43 | 0.41 | 0.35 | 0.17 | 0.20 | 0.28 | 0.26 | 0.24 | 0.44 |
| | VMI | 2.60 | 3.93 | 6.27 | 4.42 | 4.25 | 0.82 | 0.41 | 0.36 | 0.30 | 0.14 | 0.24 | 0.25 | 0.18 | 0.16 | 0.34 |
| | Jitter | 3.08 | 5.43 | 8.32 | 8.78 | 9.60 | 0.99 | 0.88 | 0.66 | 0.77 | 0.52 | 0.20 | 0.26 | 0.22 | 0.35 | 0.50 |
| ROME | Clean | 33.25 | 39.78 | 47.94 | 54.08 | 60.21 | 13.76 | 23.08 | 34.73 | 40.57 | 46.41 | 3.36 | 8.49 | 14.90 | 20.36 | 25.82 |
| | FGSM | 13.52 | 7.75 | 6.52 | 10.86 | 9.65 | 4.73 | 4.87 | 4.62 | 3.63 | 3.04 | 0.78 | 0.58 | 0.78 | 0.69 | 0.38 |
| | PGD | 12.86 | 3.61 | 3.57 | 5.87 | 3.74 | 3.29 | 2.16 | 1.76 | 0.95 | 0.96 | 0.42 | 0.15 | 0.14 | 0.09 | 0.07 |
| | CW | 13.73 | 5.62 | 4.57 | 5.51 | 4.00 | 3.70 | 3.09 | 2.51 | 2.71 | 1.58 | 0.58 | 0.56 | 0.09 | 0.03 | 0.03 |
| | VMI | 12.83 | 4.87 | 2.27 | 6.53 | 4.62 | 3.21 | 2.57 | 2.00 | 1.45 | 0.94 | 0.98 | 0.12 | 0.30 | 0.12 | 0.02 |
| | Jitter | 13.29 | 10.51 | 9.73 | 11.43 | 10.74 | 3.19 | 2.04 | 1.78 | 1.41 | 1.03 | 0.39 | 0.17 | 0.06 | 0.04 | 0.04 |
| $C^2R$ | Clean | 35.04 | 40.94 | 48.31 | 56.52 | 64.73 | 14.33 | 24.46 | 37.12 | 42.69 | 48.25 | 5.38 | 10.99 | 18.00 | 22.77 | 27.53 |
| | FGSM | 14.79(+1.27) | 8.86(+1.11) | 10.81(+1.76) | 12.26(+1.40) | 10.65(+1.00) | 5.57(+0.84) | 5.63(+0.76) | 5.12(+0.50) | 3.99(+0.36) | 3.99(+0.95) | 0.85(+0.07) | 0.60(+0.02) | 0.80(+0.02) | 1.26(+0.08) | 2.00(+0.17) |
| | PGD | 14.26(+1.40) | 4.97(+1.35) | 6.84(+0.98) | 7.36(+1.49) | 4.93(+1.09) | 3.94(+0.65) | 3.02(+0.86) | 2.46(+0.70) | 1.83(+0.88) | 1.63(+0.67) | 0.56(+0.14) | 0.16(+0.00) | 0.19(+0.03) | 0.15(+0.02) | 0.09(+0.00) |
| | CW | 14.85(+1.12) | 6.67(+1.05) | 6.71(+0.23) | 6.83(+1.32) | 5.18(+0.92) | 4.22(+0.52) | 3.49(+0.40) | 2.95(+0.44) | 3.64(+0.93) | 2.30(+0.72) | 0.70(+0.12) | 0.59(+0.03) | 0.24(+0.00) | 0.06(+0.00) | 0.50(+0.06) |
| | VMI | 14.32(+1.49) | 6.19(+1.32) | 3.55(+0.00) | 7.87(+1.34) | 6.04(+1.42) | 4.03(+0.82) | 2.99(+0.42) | 2.27(+0.27) | 1.93(+0.48) | 1.39(+0.45) | 1.02(+0.04) | 0.31(+0.06) | 0.48(+0.18) | 0.18(+0.02) | 0.45(+0.11) |
| | Jitter | 14.49(+1.20) | 11.97(+1.46) | 10.96(+1.23) | 12.56(+1.13) | 11.86(+1.12) | 3.85(+0.66) | 2.48(+0.44) | 2.46(+0.68) | 2.32(+0.91) | 1.57(+0.54) | 0.43(+0.04) | 0.37(+0.11) | 0.16(+0.00) | 0.06(+0.00) | 0.05(+0.00) |

*Table 6.* **Test accuracy (%) of baseline methods and $C^2R$** on CIFAR-10, CIFAR-100, and Tiny-ImageNet with IPC = 1, 5, 10, 30, 50, evaluated under adversarial attacks with $|\varepsilon| = 8/255$. Relative improvements over the second-best method in each setting are shown as subscripts highlighted in blue. All results are averaged over five independent runs.

| Methods | Attack | CIFAR-10 | | | | | CIFAR-100 | | | | | Tiny-ImageNet | | | | |
|---|---|---|---|---|---|---|---|---|---|---|---|---|---|---|---|---|
| | | 1 | 5 | 10 | 30 | 50 | 1 | 5 | 10 | 30 | 50 | 1 | 5 | 10 | 30 | 50 |
| $D^4M$ | Clean | 23.39 | 42.34 | 48.16 | 63.53 | 69.81 | 11.27 | 31.90 | 40.12 | 48.15 | 51.10 | 2.16 | 6.94 | 14.25 | 35.80 | 42.89 |
| | FGSM | 0.63 | 1.05 | 1.75 | 0.82 | 0.86 | 0.18 | 0.08 | 0.16 | 0.22 | 0.23 | 0.16 | 0.18 | 0.19 | 0.38 | 0.59 |
| | PGD | 0.07 | 0.18 | 0.33 | 0.10 | 0.06 | 0.05 | 0.01 | 0.00 | 0.00 | 0.00 | 0.07 | 0.05 | 0.01 | 0.00 | 0.02 |
| | CW | 0.15 | 0.18 | 0.34 | 0.11 | 0.08 | 0.04 | 0.01 | 0.00 | 0.00 | 0.00 | 0.09 | 0.06 | 0.00 | 0.00 | 0.02 |
| | VMI | 0.14 | 0.20 | 0.42 | 0.13 | 0.08 | 0.05 | 0.01 | 0.00 | 0.00 | 0.00 | 0.09 | 0.05 | 0.01 | 0.00 | 0.02 |
| | Jitter | 1.35 | 2.88 | 3.86 | 5.23 | 5.96 | 0.07 | 0.06 | 0.08 | 0.21 | 0.21 | 0.07 | 0.05 | 0.04 | 0.11 | 0.15 |
| ROME | Clean | 33.25 | 39.78 | 47.94 | 54.08 | 60.21 | 13.76 | 23.08 | 34.73 | 40.57 | 46.41 | 3.36 | 8.49 | 14.90 | 20.36 | 25.82 |
| | FGSM | 7.12 | 2.58 | 1.12 | 0.89 | 0.68 | 1.22 | 0.95 | 0.83 | 0.79 | 0.35 | 0.23 | 0.02 | 0.02 | 0.01 | 0.01 |
| | PGD | 3.97 | 1.17 | 0.82 | 1.53 | 2.18 | 1.00 | 0.29 | 0.75 | 0.75 | 0.67 | 0.10 | 0.00 | 0.01 | 0.02 | 0.00 |
| | CW | 6.18 | 1.06 | 0.44 | 0.20 | 0.10 | 1.01 | 0.65 | 0.22 | 0.09 | 0.00 | 0.03 | 0.00 | 0.01 | 0.02 | 0.00 |
| | VMI | 4.24 | 1.03 | 0.31 | 0.10 | 0.10 | 0.73 | 0.21 | 0.13 | 0.03 | 0.04 | 0.07 | 0.00 | 0.00 | 0.00 | 0.00 |
| | Jitter | 8.90 | 7.31 | 6.42 | 8.42 | 8.59 | 0.79 | 0.52 | 0.48 | 0.28 | 0.28 | 0.04 | 0.00 | 0.00 | 0.02 | 0.01 |
| $C^2R$ | Clean | 35.04 | 40.94 | 48.31 | 56.52 | 64.73 | 14.33 | 24.46 | 37.12 | 42.69 | 48.25 | 5.38 | 10.99 | 18.00 | 22.77 | 27.53 |
| | FGSM | 8.32(+1.20) | 2.83(+0.25) | 1.80(+0.05) | 0.95(+0.06) | 0.91(+0.05) | 1.40(+0.18) | 1.00(+0.05) | 0.88(+0.05) | 0.85(+0.06) | 0.42(+0.07) | 0.26(+0.03) | 0.20(+0.02) | 0.14(+0.00) | 0.23(+0.00) | 0.42(+0.00) |
| | PGD | 4.17(+0.20) | 1.45(+0.28) | 0.88(+0.06) | 1.88(+0.35) | 2.58(+0.40) | 1.19(+0.19) | 0.33(+0.04) | 1.01(+0.26) | 0.81(+0.06) | 0.70(+0.03) | 0.13(+0.03) | 0.10(+0.05) | 0.04(+0.03) | 0.06(+0.04) | 0.07(+0.05) |
| | CW | 6.28(+0.10) | 1.24(+0.18) | 0.49(+0.05) | 0.26(+0.06) | 0.17(+0.07) | 1.16(+0.15) | 0.65(+0.00) | 0.27(+0.05) | 0.13(+0.04) | 0.03(+0.03) | 0.14(+0.05) | 0.05(+0.00) | 0.01(+0.01) | 0.06(+0.04) | 0.02(+0.00) |
| | VMI | 5.47(+1.23) | 1.28(+0.25) | 0.46(+0.04) | 0.10(+0.00) | 0.10(+0.00) | 0.76(+0.03) | 0.22(+0.01) | 0.18(+0.05) | 0.07(+0.04) | 0.00(+0.00) | 0.14(+0.05) | 0.07(+0.02) | 0.00(+0.00) | 0.02(+0.02) | 0.03(+0.01) |
| | Jitter | 10.10(+1.20) | 8.44(+1.13) | 7.32(+0.90) | 9.90(+1.48) | 10.17(+1.58) | 0.75(+0.00) | 0.56(+0.04) | 0.48(+0.00) | 0.34(+0.06) | 0.35(+0.07) | 0.09(+0.02) | 0.08(+0.03) | 0.05(+0.01) | 0.06(+0.00) | 0.16(+0.01) |

*Table 7.* **Test accuracy (%) of baseline methods and $C^2R$** after AutoAttack attack on CIFAR-10, CIFAR-100, and Tiny-ImageNet with IPC = 1, 5, 10, 30, 50. In the second column, the perturbation budget is reported in units normalized by 255. Relative improvements over the second-best method in each setting are shown as subscripts highlighted in blue. All results are averaged over five independent runs.

| Methods | Attack | CIFAR-10 | | | | | CIFAR-100 | | | | | Tiny-ImageNet | | | | |
|---|---|---|---|---|---|---|---|---|---|---|---|---|---|---|---|---|
| | | 1 | 5 | 10 | 30 | 50 | 1 | 5 | 10 | 30 | 50 | 1 | 5 | 10 | 30 | 50 |
| $D^4M$ | Clean | 23.39 | 42.34 | 48.16 | 63.53 | 69.81 | 11.27 | 31.90 | 40.12 | 48.15 | 51.10 | 2.16 | 6.94 | 14.25 | 35.80 | 42.89 |
| | 1 | 14.75 | 26.11 | 32.19 | 39.75 | 43.00 | 5.60 | 12.21 | 13.76 | 14.94 | 13.49 | 0.82 | 1.72 | 2.63 | 6.70 | 10.49 |
| | 2 | 7.88 | 14.10 | 18.51 | 20.96 | 21.77 | 2.54 | 3.79 | 3.70 | 3.45 | 2.56 | 0.46 | 0.77 | 0.72 | 1.31 | 2.13 |
| | 3 | 4.24 | 6.75 | 10.23 | 9.48 | 9.45 | 1.22 | 1.11 | 0.91 | 0.85 | 0.48 | 0.26 | 0.35 | 0.22 | 0.34 | 0.53 |
| | 4 | 1.93 | 3.06 | 5.07 | 3.58 | 3.50 | 0.63 | 0.29 | 0.31 | 0.23 | 0.12 | 0.18 | 0.17 | 0.11 | 0.09 | 0.24 |
| ROME | Clean | 33.25 | 39.78 | 47.94 | 54.08 | 60.21 | 13.76 | 23.08 | 34.73 | 40.57 | 46.41 | 3.36 | 8.49 | 14.90 | 20.36 | 25.82 |
| | 1 | 22.22 | 25.42 | 28.89 | 36.92 | 38.73 | 8.87 | 12.26 | 14.78 | 16.73 | 16.73 | 1.98 | 2.89 | 4.57 | 4.21 | 3.88 |
| | 2 | 19.37 | 15.85 | 16.10 | 20.09 | 18.99 | 5.97 | 6.78 | 6.69 | 5.99 | 5.53 | 1.00 | 0.82 | 1.05 | 1.97 | 0.49 |
| | 3 | 15.62 | 7.41 | 5.84 | 10.94 | 8.29 | 4.26 | 1.85 | 2.02 | 1.74 | 1.71 | 0.48 | 0.12 | 0.18 | 0.14 | 0.01 |
| | 4 | 12.24 | 3.85 | 2.03 | 4.94 | 3.10 | 2.43 | 1.53 | 1.77 | 0.86 | 0.82 | 0.25 | 0.04 | 0.07 | 0.00 | 0.03 |
| $C^2R$ | Clean | 35.04 | 40.94 | 48.31 | 56.52 | 64.73 | 14.33 | 24.46 | 37.12 | 42.69 | 48.25 | 5.38 | 10.99 | 18.00 | 22.77 | 27.53 |
| | 1 | 24.10(+1.88) | 26.94(+0.83) | 30.56(+0.00) | 39.95(+0.20) | 40.67(+0.00) | 9.55(+0.68) | 13.71(+1.45) | 15.82(+1.04) | 18.48(+1.75) | 18.25(+1.52) | 2.20(+0.22) | 3.40(+0.51) | 4.60(+0.03) | 4.41(+0.00) | 4.53(+0.00) |
| | 2 | 21.20(+1.83) | 17.48(+1.63) | 18.95(+0.44) | 22.08(+1.12) | 22.49(+0.72) | 6.78(+0.81) | 7.48(+0.70) | 7.03(+0.34) | 6.15(+0.16) | 6.49(+0.96) | 1.34(+0.34) | 0.91(+0.09) | 1.15(+0.10) | 2.82(+0.85) | 1.09(+0.00) |
| | 3 | 17.60(+1.98) | 8.14(+0.73) | 6.38(+0.00) | 12.43(+1.49) | 8.67(+0.00) | 4.81(+0.55) | 2.68(+0.83) | 2.64(+0.62) | 2.60(+0.86) | 2.29(+0.58) | 1.08(+0.60) | 0.17(+0.00) | 0.29(+0.07) | 0.20(+0.00) | 0.62(+0.09) |
| | 4 | 13.36(+1.12) | 3.95(+0.10) | 2.31(+0.00) | 5.58(+0.64) | 3.56(+0.06) | 2.80(+0.37) | 1.74(+0.21) | 2.04(+0.27) | 1.80(+0.94) | 1.47(+0.65) | 0.56(+0.31) | 0.22(+0.05) | 0.18(+0.07) | 0.12(+0.03) | 0.35(+0.11) |

*Table 8.* **Test accuracies (%) on different datasets for baseline methods and C²R** with IPC = 1, 10, 50 on ImageNet-1K subsets, evaluated under adversarial attacks with $|\varepsilon| = 4/255$. Each result is averaged over five independent runs.

| Method | | ImageNette | | | ImageWoof | | | ImageFruit | | | ImageMeow | | | ImageSquawk | | | ImageYellow | | |
|---|---|---|---|---|---|---|---|---|---|---|---|---|---|---|---|---|---|---|---|
| | | 1 | 10 | 50 | 1 | 10 | 50 | 1 | 10 | 50 | 1 | 10 | 50 | 1 | 10 | 50 | 1 | 10 | 50 |
| MTT | Clean | 48.20 | 66.40 | 67.60 | 30.40 | 38.00 | 39.40 | 25.00 | 42.20 | 44.60 | 31.00 | 44.40 | 44.20 | 39.00 | 55.60 | 59.20 | 44.60 | 63.40 | 66.20 |
| | FGSM | 7.40 | 10.80 | 8.40 | 2.60 | 0.80 | 0.00 | 2.80 | 2.40 | 2.20 | 2.00 | 1.40 | 0.80 | 4.20 | 4.40 | 3.80 | 6.40 | 9.40 | 9.00 |
| | PGD | 5.40 | 4.60 | 2.60 | 2.00 | 0.20 | 0.00 | 1.80 | 0.40 | 1.00 | 0.60 | 0.80 | 0.60 | 1.80 | 2.40 | 2.00 | 3.00 | 5.00 | 4.60 |
| | CW | 4.80 | 4.60 | 1.40 | 1.40 | 0.40 | 0.00 | 1.60 | 0.60 | 1.00 | 0.20 | 0.60 | 0.60 | 2.20 | 2.60 | 1.60 | 2.60 | 5.60 | 3.80 |
| | VMI | 5.60 | 5.40 | 2.00 | 2.00 | 0.40 | 0.00 | 1.80 | 0.60 | 1.00 | 0.80 | 0.80 | 0.60 | 2.40 | 2.80 | 1.60 | 4.00 | 5.80 | 3.80 |
| | Jitter | 8.00 | 12.20 | 13.00 | 2.20 | 2.40 | 0.40 | 2.60 | 3.20 | 3.60 | 1.60 | 2.20 | 2.60 | 5.60 | 6.40 | 7.00 | 7.40 | 11.40 | 10.40 |
| C²R | Clean | 44.10 | 63.80 | 64.70 | 28.50 | 36.30 | 37.40 | 22.80 | 39.60 | 44.20 | 27.50 | 44.20 | 44.60 | 35.00 | 54.20 | 58.90 | 42.70 | 60.30 | 63.80 |
| | FGSM | 8.30 | 11.30 | 9.30 | 3.00 | 1.00 | 0.00 | 2.90 | 3.00 | 2.00 | 2.10 | 1.50 | 1.00 | 4.60 | 4.90 | 4.20 | 7.20 | 10.40 | 9.70 |
| | PGD | 6.20 | 5.10 | 3.00 | 2.50 | 0.50 | 0.00 | 2.20 | 0.40 | 1.10 | 0.80 | 0.80 | 0.70 | 1.90 | 2.60 | 2.40 | 3.20 | 5.70 | 4.70 |
| | CW | 5.00 | 5.20 | 1.80 | 1.80 | 0.50 | 0.00 | 2.00 | 0.70 | 1.20 | 0.80 | 1.00 | 0.90 | 2.60 | 3.10 | 2.10 | 2.70 | 6.10 | 4.00 |
| | VMI | 6.20 | 6.00 | 2.60 | 2.50 | 0.60 | 0.00 | 2.20 | 0.80 | 1.50 | 1.10 | 1.00 | 0.70 | 2.70 | 3.00 | 2.00 | 4.50 | 6.50 | 3.90 |
| | Jitter | 9.00 | 13.00 | 13.50 | 2.20 | 2.50 | 0.30 | 3.10 | 3.50 | 3.60 | 1.80 | 2.80 | 2.70 | 6.60 | 7.00 | 7.30 | 8.30 | 12.20 | 11.20 |

*Table 9.* **Test accuracies (%) on different datasets for baseline methods and C²R** with IPC = 1, 10, 50 on ImageNet-1K subsets, evaluated under adversarial attacks with $|\varepsilon| = 8/255$. Each result is averaged over five independent runs.

| Method | | ImageNette | | | ImageWoof | | | ImageFruit | | | ImageMeow | | | ImageSquawk | | | ImageYellow | | |
|---|---|---|---|---|---|---|---|---|---|---|---|---|---|---|---|---|---|---|---|
| | | 1 | 10 | 50 | 1 | 10 | 50 | 1 | 10 | 50 | 1 | 10 | 50 | 1 | 10 | 50 | 1 | 10 | 50 |
| MTT | Clean | 48.20 | 66.40 | 67.60 | 30.40 | 38.00 | 39.40 | 25.00 | 42.20 | 44.60 | 31.00 | 44.40 | 44.20 | 39.00 | 55.60 | 59.20 | 44.60 | 63.40 | 66.20 |
| | FGSM | 1.20 | 0.80 | 1.80 | 0.60 | 0.00 | 0.00 | 0.80 | 0.20 | 0.40 | 0.00 | 0.40 | 0.20 | 0.40 | 0.60 | 1.00 | 0.60 | 2.00 | 1.80 |
| | PGD | 0.20 | 0.20 | 1.20 | 0.40 | 0.00 | 0.00 | 0.00 | 0.00 | 0.20 | 0.00 | 0.20 | 0.20 | 0.00 | 0.00 | 1.40 | 0.00 | 0.80 | 2.20 |
| | CW | 0.20 | 0.20 | 0.20 | 0.20 | 0.00 | 0.00 | 0.00 | 0.00 | 0.00 | 0.00 | 0.00 | 0.00 | 0.20 | 0.20 | 0.20 | 0.00 | 0.40 | 0.00 |
| | VMI | 0.40 | 0.20 | 0.20 | 0.40 | 0.00 | 0.00 | 0.00 | 0.00 | 0.00 | 0.00 | 0.00 | 0.00 | 0.00 | 0.20 | 0.20 | 0.00 | 1.00 | 0.00 |
| | Jitter | 6.60 | 11.40 | 9.80 | 1.80 | 2.60 | 2.00 | 2.20 | 2.40 | 3.20 | 2.60 | 1.60 | 2.40 | 3.80 | 5.40 | 4.60 | 6.00 | 9.80 | 8.80 |
| C²R | Clean | 44.10 | 63.80 | 64.70 | 28.50 | 36.30 | 37.40 | 22.80 | 39.60 | 44.20 | 27.50 | 44.20 | 44.60 | 35.00 | 54.20 | 58.90 | 42.70 | 60.30 | 63.80 |
| | FGSM | 1.30 | 0.90 | 2.10 | 0.80 | 0.00 | 0.00 | 1.00 | 0.40 | 0.50 | 0.00 | 0.50 | 0.60 | 0.50 | 0.70 | 1.10 | 0.80 | 2.20 | 1.90 |
| | PGD | 0.40 | 0.60 | 1.40 | 0.80 | 0.30 | 0.20 | 0.00 | 0.00 | 0.50 | 0.10 | 0.30 | 0.40 | 0.00 | 0.00 | 1.80 | 0.90 | 1.10 | 2.50 |
| | CW | 0.40 | 0.40 | 0.50 | 0.30 | 0.00 | 0.10 | 0.10 | 0.10 | 0.00 | 0.00 | 0.20 | 0.00 | 0.30 | 0.60 | 0.30 | 0.00 | 0.70 | 0.00 |
| | VMI | 0.70 | 0.50 | 0.30 | 0.80 | 0.10 | 0.10 | 0.00 | 0.30 | 0.20 | 0.10 | 0.00 | 0.20 | 0.50 | 0.30 | 0.60 | 0.60 | 1.10 | 0.30 |
| | Jitter | 7.50 | 12.10 | 10.80 | 2.10 | 2.90 | 2.20 | 2.80 | 2.50 | 3.30 | 3.20 | 1.80 | 2.70 | 3.90 | 6.00 | 4.70 | 6.80 | 10.50 | 9.60 |

## C.2. Robustness Degradation under Adversarial Perturbations

To additionally evaluate robustness degradation under FGSM perturbations, we measure the drop rate of baseline methods and C²R across all datasets and IPC settings. As shown in Fig. 7, C²R achieves the lowest drop rate on CIFAR-10, CIFAR-100, Tiny-ImageNet, and ImageNette, and its curve remains consistently below those of competing methods as the synthetic dataset scales. We observe that several baselines incur substantial degradation under FGSM, whereas C²R maintains a noticeably flatter trajectory, suggesting that its margin-oriented design reduces vulnerability to gradient-based perturbations and preserves more stable accuracy. These results indicate that C²R improves robustness in the FGSM regime as well, further confirming the stability of its distilled representations.

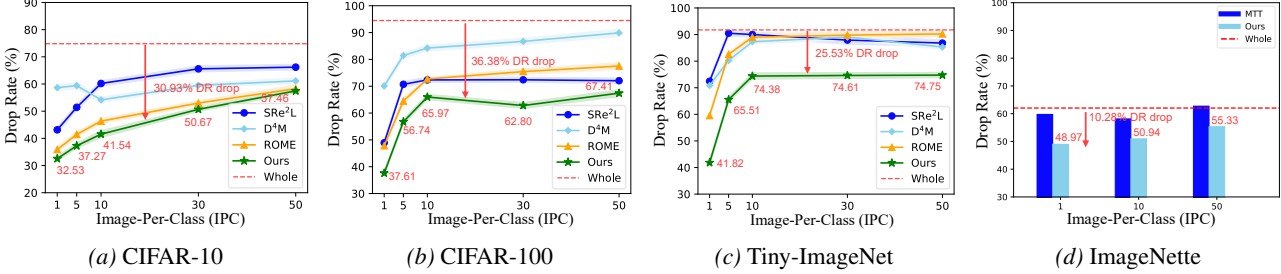

| *(a)* CIFAR-10 | *(b)* CIFAR-100 | *(c)* Tiny-ImageNet | *(d)* ImageNette |

*Figure 7.* **Drop rate (DR) comparison under FGSM attacks across datasets and IPC.** The red dashed line denotes the DR of models trained on the whole dataset. C²R consistently achieves the lowest drop rate and exhibits a flatter trajectory as the synthetic set scales.

## C.3. Comparisons with RDD

To evaluate the behavior of distilled datasets under distributional shifts, we compare $C^2R$ with RDD (Vahidian et al., 2025) following the experimental settings in the RDD. As reported in Tab. 10, $C^2R$ achieves higher accuracy than RDD across most settings on both datasets, while maintaining comparable performance under clean evaluation. We observe that RDD mitigates part of the distributional gap through group-level modeling yet remains sensitive to corruptions that alter class-level geometry, whereas the margin-oriented design of $C^2R$ preserves more stable decision boundaries and reduces degradation under feature-level perturbations. These results indicate that $C^2R$ transfers more reliably under distribution shifts and provides improved robustness in scenarios beyond the main adversarial threat model.

*Table 10.* **Test accuracies (%) on different datasets for RDD and $C^2R$** with IPC $= 10$ on CIFAR-10 and ImageNet-10. Each result is averaged over five independent runs.

| Method | CIFAR-10 | | ImageNet-10 | |
|---|---|---|---|---|
| | RDD | $C^2R$ | RDD | $C^2R$ |
| **Clean** | 60.2 | 60.4 | 55.2 | 54.6 |
| Cluster-min | 46.7 | 48.5 | 47.1 | 48.9 |
| Noise | 49.5 | 52.6 | 53.9 | 54.8 |
| Blur | 39.0 | 42.2 | 54.6 | 57.0 |
| Invert | 13.0 | 14.1 | 21.6 | 24.0 |

# D. Limitations and Future Work

One limitation in this work is that we focus on image classification benchmarks with moderate resolutions (CIFAR-10/100, Tiny-ImageNet, ImageNet subsets) and standard architectures; it remains unclear how well $C^2R$ transfers to higher-resolution datasets, dense prediction tasks, or more diverse backbone families. In future work, we plan to extend $C^2R$ to large-scale and high-resolution settings vision tasks, investigating whether robust curricula and contrastive alignment can benefit detection, graph and continual learning.

# E. More Clarifications of the Smallest Robust Margin.

Our method is margin-centric in an optimization sense rather than a measurement claim. In AAC, the robust-hinge surrogate is $h_{\mathrm{rob}}(x, y; \theta) = [1 - m_{\mathrm{rob}}(x, y; \theta)]_+$; AAC is equivalent to optimizing a tail-dominated robust risk $\sum_i w_i\, h_{\mathrm{rob}}(x_i, y_i; \theta)$ with nondecreasing weights $w_i \propto \phi(s_i)$, thereby concentrating gradient updates on the largest hinges (smallest robust margins) rather than uniformly averaging over all adversarial samples. In the limiting case where the weighting becomes a hard top-$q$ selector and $q \to 1$, this objective approaches $\max_i h_{\mathrm{rob}}(x_i, y_i; \theta)$, which is a direct surrogate for maximizing $\min_i m_{\mathrm{rob}}(x_i, y_i; \theta)$. Hence, the smallest robust margin can be interpreted as targeting its tail surrogate via attack-aware reweighting. As future work, we will further provide dedicated empirical analyses (e.g., robust-margin statistics and tail distributions) to validate these margin-centric claims more directly.

