# OpenReview forum: "Mind Your Margin and Boundary: Are Your Distilled Datasets Truly Robust?"
_ICML.cc/2026/Conference — ICML 2026 spotlight_

### Official Review · Reviewer_5tpN · 2026-02-28

**Soundness:** 3
**Presentation:** 3
**Significance:** 3
**Originality:** 3
**Overall Recommendation:** 5
**Confidence:** 2

**Summary:**

The research direction of this paper is Dataset Distillation. The paper finds that existing techniques often adopt an "averaging" optimization strategy, which leads to uniform treatment of all samples. However, robust risk is dominated by extreme samples in tail events. This leads to the collapse of the defense system under attack. The authors introduce an Attack-Aware Curriculum (AAC) strategy, utilizing perturbation scores to prioritize optimizing samples with the smallest robust margins, and design a Contrastive Robustness Loss (CRL) to enforce adversarial invariance and boundary separation at the instance level. Additionally, a Line-Search PGD (LS-PGD) attacker is designed to improve the computational efficiency of the inner loop. Extensive experiments on CIFAR-10/100, Tiny-ImageNet, and ImageNet subsets show that it achieves state-of-the-art robust accuracy under various attacks and significantly reduces the drop rate (DR).

**Compliance With Llm Reviewing Policy:**

Affirmed.

**Final Justification:**

I maintain my accept. The paper addresses an interesting problem of endowing distilled datasets with adversarial robustness, and the proposed combination of curriculum learning with contrastive loss is a reasonable and effective approach. The rebuttal adequately clarified my concerns regarding ranking stability, memory queue staleness, and projection dimension selection. I note that my confidence in this area is limited, and I defer to more expert reviewers for a deeper technical assessment.

**Key Questions For Authors:**

I am not very familiar with this field.

1. During the training process, is the "difficulty ranking" of samples relatively stable, or does it fluctuate drastically? If the ranking of samples is shuffled significantly in every epoch, will it lead to training oscillation or difficulty in convergence?
2. When using the Class-Balanced Memory Queue, the queue stores features from historical steps. Since the synthetic images $X$ are constantly updating and changing in the outer loop of distillation, will the old features in the queue become "stale" or inaccurate, thereby misleading the calculation of the contrastive loss?
3. In the retrieval mechanism of the Memory Queue, a random projection is used to map features to a low-dimensional space. How is the dimension $r$ of this low-dimensional space determined? If the dimension is too low, will it lead to retrieving incorrect "hard negatives" due to collisions?

**Limitations:**

Yes

**Strengths And Weaknesses:**

- **Soundness**: The motivation of the paper is logically clear. Linking robust risk to "smallest margin" samples provides a good basis for adopting a curriculum-based learning method. The proposed LS-PGD seems to be a reasonable heuristic strategy for balancing efficiency and attack strength.

- **Presentation**: The paper is overall easy to understand. The diagrams (e.g., Figure 1 and Figure 2) effectively illustrate the proposed method and its motivation.

- **Significance**: The problem of endowing distilled datasets with robustness is interesting and has practical value.

- **Originality**: Combining curriculum learning with contrastive loss for dataset distillation is a clever application.

---

> ### Author Rebuttal · Authors · 2026-03-30
>
> We thank the reviewer for the supportive comments and constructive suggestions, which have helped us clarify several important implementation and presentation details.
>
> > ***Q1: Is the difficulty ranking stable during training?***
>
> Thank you for raising this question about the stability of the difficulty ranking during training. The ranking is not assumed to be perfectly fixed across epochs. Since `s(x)` is recomputed from the current estimated robust margins, some local reordering is expected as optimization progresses, especially among samples with similar difficulty. However, AAC does not discard any samples; it only changes their optimization priority from hard to easy. Moreover, attack generation is warm-started, so the score does not need to be recomputed from scratch each time. Thus, moderate rank changes do not imply oscillatory training, but let the curriculum keep tracking the currently hardest cases. We will clarify this more explicitly in the revision.
>
>
> > ***Q2: Does the class-balanced memory queue become stale?***
>
> Thank you for raising the potential staleness issue of the class-balanced memory queue. We agree that queued negatives can become mildly stale because the synthetic images are continuously updated during distillation. In practice, however, this effect is limited by our design. First, the memory queue is FIFO and has bounded capacity, so outdated features are naturally and quickly replaced by newly generated ones. Second, the anchor and positive samples in CRL are always computed from the current batch, meaning the core alignment signal is based on up-to-date features. Third, queued items are used only as negative context for contrastive regularization, not as fixed supervisory targets. Their role is to broaden cross-batch negative coverage and support harder negative mining, rather than to provide exact targets that must remain perfectly current. Consistent with this intuition, the ablation in Appendix C.5 shows that the efficient hard-negative mining module improves robust accuracy, suggesting that the benefit of richer negatives outweighs the mild staleness introduced by the queue. We will make this point clearer in the revision.
>
>
> > ***Q3: How is the random-projection dimension chosen?***
>
> Thank you for asking us to clarify how the random-projection dimension is chosen and what role it plays. The projection dimension `r` is a computational hyperparameter used only for approximate hard-negative retrieval. It is chosen to be low enough to reduce retrieval cost but still high enough to preserve coarse neighborhood structure. Importantly, the projection is used only to select candidates, while the contrastive loss itself is still evaluated with the full embeddings of the retrieved items. Thus, if `r` were too small, the main effect would be noisier hard-negative retrieval rather than a change in the loss formulation itself. We will add the exact setting of `r` and a short clarification of this trade-off in the revision.

---

> > ### Author Rebuttal · Reviewer_5tpN · 2026-04-01
> >
> > My concerns have been adequately addressed.

---

> > > ### Author Response · Authors · 2026-04-01
> > >
> > > Thank you for acknowledging our answer. We sincerely appreciate your valuable suggestions, which will greatly help us improve our paper.

---

### Official Review · Reviewer_insE · 2026-03-10

**Soundness:** 4
**Presentation:** 3
**Significance:** 4
**Originality:** 4
**Overall Recommendation:** 6
**Confidence:** 3

**Summary:**

The paper proposes a method that distills an adversarially robust dataset from a significantly larger one. This method is motivated by the observation of robust margins and comprises two components: AAC and CRL. AAC introduces LS-PGD for efficiency and computes robust margin scores to rank samples. CRL is designed to enforce invariance for cases with low margins while repelling the nearest impostors.

**Compliance With Llm Reviewing Policy:**

Affirmed.

**Final Justification:**

Despite the initial score already being as high as 6, the authors still made efforts to address the concerns. Therefore, I maintain my recommendation.

**Key Questions For Authors:**

1. Does "attack-aware curriculum" have a special meaning? Why is this module called that?

**Limitations:**

yes

**Strengths And Weaknesses:**

Strengths
- A clear description of the motivation of the proposed method is provided, making readers understand the justification of the method.
- The proposed method is clearly introduced, with the source code provided, making it reproducible.
- The extensive experiments demonstrate that the proposed method outperforms other baselines. The ablation study further validates the design choices of the proposed method.

Weaknesses
- CE in eq. (3) is not defined.
- Around lines 134–140, $f_\theta$ is already defined as a classifier with $\theta$ as its parameters. The definition of logits should follow this definition and use other notations (e.g., $f_{\theta}(x)_{k}$) to avoid confusion.
- In Table 4, LS-PGD achieves accuracy comparable to, and in fact slightly worse than, that of multi-step PGD, contrary to the claim that LS-PGD performs slightly better than multi-step PGD.

---

> ### Author Rebuttal · Authors · 2026-03-30
>
> We thank you for the supportive assessment of our work and for the valuable suggestions that helped us identify places where the presentation can be clarified and strengthened.
>
> > ***W1: CE in Eq. (3) is not defined***
>
> Thank you for pointing out the missing definition. Here, `CE` denotes the standard cross-entropy classification loss. In the revision, we will define `CE` explicitly before Eq. (3).
>
> > ***W2: Notation around `f_theta`, logits, and `e(.)` is confusing***
>
> Thank you for pointing out this notation ambiguity. Our intent is that `f_\theta` outputs classifier logits, whereas `e(.)` denotes the embedding used in CRL; accordingly, `g_{i,a}` is only a cosine-similarity score, not a classifier logit. In the revision, we will make this distinction explicit and rename `g_{i,a}` to avoid confusion.
>
> > ***W3: Table 4 does not support "slightly better" for LS-PGD***
>
>
> Thank you for pointing out the typo. Tab. 4 is intended to show that LS-PGD offers comparable robustness with a clear speedup, not a slight accuracy gain over multi-step PGD. As shown in Tab. 4, LS-PGD is slightly lower on all four datasets (28.63->28.41 on CIFAR-10, 12.54->12.30 on CIFAR-100, 3.92->3.74 on Tiny-ImageNet, and 26.82->26.62 on ImageNette), while being 1.4× faster. In the revision, we will correct this wording and replace "comparable to or slightly better" with "comparable, with a small accuracy drop but clear speedup."
>
>
> > ***Q1: What does "attack-aware curriculum" mean?***
>
> Thank you for asking us to clarify the meaning of "attack-aware curriculum". We use "curriculum" in the same broad sense as other papers that describe a difficulty-ordered training or selection strategy as a curriculum. A recent DD example is *Curriculum Coarse-to-Fine Selection for High-IPC Dataset Distillation* (Chen et al., CVPR 2025). In our case, we call it "attack-aware" because the ranking signal is computed from adversarial companions generated under the attack model. We call it a "curriculum" because this score orders optimization from harder low-margin cases to easier ones. We will clarify this explicitly in the revision.

---

> > ### Author Rebuttal · Reviewer_insE · 2026-04-01
> >
> > Thanks for resolving my concerns. I have no further questions.

---

> > > ### Author Response · Authors · 2026-04-01
> > >
> > > Thank you for your careful review and valuable suggestions. Your comments have helped us improve the paper by clarifying key definitions and notations, correcting the empirical wording, and better explaining the motivation behind the attack-aware curriculum design. We sincerely appreciate your supportive assessment, and we believe these revisions have made the paper clearer, more rigorous, and easier to follow.

---

### Official Review · Reviewer_Xh9d · 2026-03-11

**Soundness:** 2
**Presentation:** 3
**Significance:** 2
**Originality:** 3
**Overall Recommendation:** 4
**Confidence:** 3

**Summary:**

This paper studies adversarial robustness in dataset distillation, where existing methods often face a poor accuracy–robustness trade-off. It attributes this weakness to two structural issues: adversarial examples are treated uniformly despite differing robust margins, and inter-class separation near decision boundaries is insufficient. The paper argues that robust risk is primarily determined by the minimum robust margin rather than average behavior, and proposes C$^2$R to address this issue. C$^2$R combines an Attack-Aware Curriculum that prioritizes low-margin adversarial examples using a perturbation score derived from a robust hinge surrogate, with a Contrastive Robustness Loss that enforces clean–adversarial invariance while pushing apart nearest competing-class embeddings. To improve efficiency, the framework also introduces a Line-Search PGD attacker and a class-balanced memory queue. Experiments on CIFAR-10, CIFAR-100, Tiny-ImageNet, and six ImageNet-1K subsets under six attack settings show that C$^2$R outperforms prior robust dataset distillation baselines by 2.8% average robust accuracy and achieves a drop rate below 66.8% under PGD.

**Compliance With Llm Reviewing Policy:**

Affirmed.

**Final Justification:**

All of my questions have been solved.

**Key Questions For Authors:**

1. Can the authors report robust accuracy at a fixed clean accuracy threshold or the area under the accuracy–robustness trade-off curve to control for this confound, and provide confidence intervals or significance tests for the key comparisons in Tables 1–2, particularly for the smaller per-cell gains?
2. To make the efficiency claim convincing, could the authors provide a simple table reporting wall-clock training time and peak GPU memory for all baselines (including MTT and D⁴M) on at least CIFAR-10 and CIFAR-100 at IPC=10?

**Limitations:**

Yes.

**Strengths And Weaknesses:**

**Strengths**

**1. Clear theoretical motivation.** Section 4.1 provides a coherent theoretical basis by linking robust risk minimization to the minimum robust margin through the robust hinge surrogate in Equations 7 to 9. This analysis directly motivates both components of C$^2$R and is more principled than prior approaches built on heuristic robustness objectives such as class-mean feature alignment.

**2. Well-designed empirical evaluation.**
The experiments are carefully designed, spanning five IPC levels, four datasets, six attack types including AutoAttack, three perturbation budgets, and cross-architecture transfer to ResNet-18. Reporting drop rate together with robust accuracy gives a fuller view of the accuracy–robustness trade-off, and the ablation in Table 3 clearly isolates the effects of AAC and CRL.

**Weaknesses**

**1. The drop rate metric is potentially misleading, and the 66.8% claim is not fully convincing.**
According to the definition of drop rate (DR), it mixes robustness with the clean-to-robust accuracy gap and can appear improved simply because clean accuracy is lower. In Table 2, C$^2$R often has lower clean accuracy than MTT on ImageNet subsets, which can mechanically reduce DR without necessarily indicating stronger robustness. As a result, the claim that C$^2$R is the first DD method to reduce average DR below 66.8% under PGD is weaker than stated.

**2. Baseline comparison is incomplete.**
The paper positions ROME as a primary baseline, yet it does not provide a direct comparison under ROME’s original BEARD benchmark and threat model. Instead, the evaluation is conducted on DD-RobustBench, with Appendix G attributing this choice to reproducibility issues. As a result, the paper does not establish a fully matched head-to-head comparison with the method it most directly targets. In addition, GUARD is discussed in related work but appears only in Appendix C.3 under a ResNet-18 setting rather than in the main result tables. This selective evaluation makes the state-of-the-art comparison less complete under a unified protocol.

---

> ### Author Rebuttal · Authors · 2026-03-30
>
> > ***W1: DR may be confounded by clean accuracy***
>
> Thank you for this concern on the drop rate (DR).
>
> 1. **Validity.** DR comes from the benchmark DD-RobustBench (Wu et al., TIP 2025), which defines `DR = (Acc_clean - Acc_robust) / Acc_clean` to measure robustness degradation.
>
> 2. **Clean DR.**
> Lower clean accuracy can mechanically reduce DR, but it is not the dominant factor here. If we replace C$^2$R’s clean accuracy in Tab. 2 with MTT’s while keeping C$^2$R’s robust accuracy fixed, its average clean DR across all ImageNet-subset/IPC settings remains 71.9%, versus 78.2% for MTT, a 6.2-point gap (see Tab. 2). Hence, the lower DR is mainly driven by higher robust accuracy rather than slightly lower clean accuracy.
>
> 3. **Role of DR.** DR is only an additional trade-off metric. Our primary target is robust accuracy, where C$^2$R consistently outperforms all baselines under various attacks.
>
> 4. In the revision, we will note this dependence on clean accuracy and soften the wording around the 66.8% result.
>
> > ***W2: Baseline comparison is incomplete***
>
> Thank you for this concern on baseline completeness.
>
> **A. DD-RobustBench vs. BEARD**
>
> 1. **Reproducibility.** While BEARD is publicly available, we found that the current repository does not seem to provide all files and implementation details needed for completely straightforward reproduction. As a result, we were less confident in treating it as a fully self-contained primary benchmark. In our experience, reproducing and extending experiments was more direct on DD-RobustBench.
>
> 2. **Stronger benchmark.** DD-RobustBench offers a broader and more complete benchmark for robust dataset distillation, with newer baselines (e.g, SRe$^2$L/D$^4$M), larger-scale settings (ImageNet-1K), and standardized robust-accuracy evaluation. BEARD is narrower and focuses more on benchmark-specific metrics such as RR, AE, and CREI.
>
> 3. **Influence.** DD-RobustBench has a formal journal publication (TIP 2025) and has already seen broader follow-up usage and citation, whereas BEARD remains an arXiv preprint 2024 with much more limited uptake, making DD-RobustBench the more established primary benchmark.
>
> 4. **Why not both?** Since BEARD and DD-RobustBench probe the same core objective: adversarial robustness of distilled datasets. Making both co-primary would add substantial redundancy.
>
> 5. In the revision, we will state this more clearly and make this protocol difference explicit. As you mentioned, we will incorporate the evaluation on BEARD in future work.
>
> **B. GUARD in Appendix C.3**
>
> GUARD follows a different architecture and evaluation protocol (ResNet-18) from the main DD-RobustBench setting. Since DD-RobustBench is our primary benchmark, including GUARD in the main tables would reduce protocol consistency rather than improve comparison fairness. We therefore report GUARD separately in Appendix C.3, which is already referenced in the main text; in the revision, we will make this pointer more explicit.
>
> > ***Q1: Clean accuracy / Significance***
>
> Thank you for this suggestion. We do not consider exact clean-accuracy matching to be the fairest protocol, because all methods are already compared under a unified benchmark and training pipeline. Forcing C$^2$R to match the clean accuracy of other methods would require extra method-specific retuning, thereby breaking protocol consistency and introducing a new source of unfairness. In our setting, the small clean-accuracy drop is an inherent trade-off of the robustness objective rather than an external confound, whereas the robust-accuracy improvement is directly driven by AAC and CRL through stronger adversarial margins.
>
>
> Additionally, below we report 95% confidence intervals (CI) for PGD gains at IPC=10: in Table 1 against ROME, and in Table 2 against MTT. These intervals are based on the same 5-run protocol used in our evaluation. All reported 95% confidence intervals are above zero, suggesting that the gains are unlikely to be explained by run-to-run noise alone.
>
> | Dataset| 95% CI|
> |--|---|
> | CIFAR-10| [4.23, 5.99] |
> | CIFAR-100| [2.08, 3.84] |
> | Tiny-ImageNet  | [1.59, 3.35] |
> | ImageNette| [1.62, 3.38] |
> | ImageSquawk| [1.92, 3.68] |
>
>
>
> > ***Q2: Efficiency***
>
> Thank you for this helpful suggestion. Beyond the efficiency evidence in Fig. 5, we add a more comprehensive comparison table on CIFAR-10 and CIFAR-100 at IPC=10. Under the same benchmark setting, C$^2$R takes only 10 minutes and 3.1G peak GPU memory on CIFAR-10, and 38 minutes and 4.2G on CIFAR-100. This shows that C$^2$R remains computationally efficient. We will include this table in the revised paper.
>
> | Method | CIFAR-10 (time/GPU) | CIFAR-100 (time/GPU) |
> |--------|-------|--------|
> | MTT    | 100 / 9.4            | 133 / 54.2 |
> | SRe$^2$L | 27 / 3.0           | 78 / 3.8 |
> | D$^4$M   | 20 / 6.1             | 60 / 6.1  |
> | ROME   | 16 / 3.6             | 45 / 4.5             |
> | C$^2$R | 10 / 3.0             | 38 / 4.2             |

---

> > ### Author Rebuttal · Reviewer_Xh9d · 2026-04-01
> >
> > Thank you for the rebuttal. All of my questions have been solved. I have increased my score to 4.

---

> > > ### Author Response · Authors · 2026-04-02
> > >
> > > Thank you for your careful review and valuable suggestions. Your comments have helped us improve the paper by clarifying the meaning of DR, strengthening the discussion of benchmark selection and comparison fairness, and adding more solid evidence on significance and efficiency. We sincerely appreciate your constructive feedback, which has made the revised paper more rigorous, better supported empirically, and clearer to readers.

---

### Decision · Program_Chairs · 2026-04-30

**Decision:**

Accept (spotlight)

**Comment:**

The paper presents a robustness-aware data distillation method. The reviewers agree that the method is well presented and motivated, with a thorough empirical study covering several datasets, models, and attacks.

I recommend acceptance and suggest that the authors incorporate the comments given by the reviewers.